# Second Order PAC-Bayesian Bounds
# for the Weighted Majority Vote

**Andrés R. Masegosa**[*]        **Stephan S. Lorenzen**        **Christian Igel**        **Yevgeny Seldin**
University of Almería                    University of Copenhagen
andresma@ual.es                    {lorenzen,igel,seldin}@di.ku.dk

## Abstract

We present a novel analysis of the expected risk of weighted majority vote in multiclass classification. The analysis takes correlation of predictions by ensemble members into account and provides a bound that is amenable to efficient minimization, which yields improved weighting for the majority vote. We also provide a specialized version of our bound for binary classification, which allows to exploit additional unlabeled data for tighter risk estimation. In experiments, we apply the bound to improve weighting of trees in random forests and show that, in contrast to the commonly used first order bound, minimization of the new bound typically does not lead to degradation of the test error of the ensemble.

## 1 Introduction

Weighted majority vote is a fundamental technique for combining predictions of multiple classifiers. In machine learning, it was proposed for neural networks by Hansen and Salomon [1990] and became popular with the works of Breiman [1996, 2001] on bagging and random forests and the work of Freund and Schapire [1996] on boosting. Zhu [2015] surveys the subsequent development of the field. Weighted majority vote is now part of the winning strategies in many machine learning competitions [e.g., Chen and Guestrin, 2016, Hoch, 2015, Puurula et al., 2014, Stallkamp et al., 2012]. Its power lies in the cancellation of errors effect [Eckhardt and Lee, 1985]: when individual classifiers perform better than a random guess and make independent errors, the errors average out and the majority vote tends to outperform the individual classifiers.

A central question in the design of a weighted majority vote is the assignment of weights to individual classifiers. This question was resolved by Berend and Kontorovich [2016] under the assumptions that the expected error rates of the classifiers are known and their errors are independent. However, neither of the two assumptions is typically satisfied in practice.

When the expected error rates are estimated based on a sample, the common way of bounding the expected error of a weighted majority vote is by twice the error of the corresponding randomized classifier [Langford and Shawe-Taylor, 2002]. A randomized classifier, a.k.a. Gibbs classifier, associated with a distribution (weights) $\rho$ over classifiers draws a single classifier at random at each prediction round according to $\rho$ and applies it to make the prediction. The error rate of the randomized classifier is bounded using PAC-Bayesian analysis [McAllester, 1998, Seeger, 2002, Langford and Shawe-Taylor, 2002]. We call this a *first order bound*. The factor 2 bound on the gap between the error of the weighted majority vote and the corresponding randomized classifier follows from the observation that an error by the weighted majority vote implies an error by at least a weighted half of the base classifiers. The bound is derived using Markov's inequality. While the PAC-Bayesian bounds for the randomized classifier are remarkably tight [Germain et al., 2009, Thiemann et al., 2017], the factor 2 gap is only tight in the worst-case, but loose in most real-life situations, where the weighted

---

[*]Part of the work was done while AM was visiting the University of Copenhagen.

majority vote typically performs better than the randomized classifier rather than twice worse. The reason for looseness is that the approach does not take the correlation of errors into account.

In order to address the weakness of the first order bound, Lacasse et al. [2007] have proposed PAC-Bayesian C-bounds, which are based on Chebyshev-Cantelli inequality (a.k.a. one-sided Chebyshev's inequality) and take correlations into account. The idea was further developed by Laviolette et al. [2011], Germain et al. [2015], and Laviolette et al. [2017]. However, the C-bounds have two severe limitations: (1) They are defined in terms of classification margin and the second moment of the margin is in the denominator of the bound. The second moment is difficult to estimate from data and significantly weakens the tightness of the bounds [Lorenzen et al., 2019]. (2) The C-bounds are difficult to optimize. Germain et al. [2015] were only able to minimize the bounds in a highly restrictive case of self-complemented sets of voters and aligned priors and posteriors. In binary classification a set of voters is self-complemented if for any hypothesis $h \in \mathcal{H}$ the mirror hypothesis $-h$, which always predicts the opposite label to the one predicted by $h$, is also in $\mathcal{H}$. A posterior $\rho$ is aligned on a prior $\pi$ if $\rho(h) + \rho(-h) = \pi(h) + \pi(-h)$ for all $h \in \mathcal{H}$. Obviously, not every hypothesis space is self-complemented and such sets can only be defined in binary, but not in multiclass classification. Furthermore, the alignment requirement only allows to shift the posterior mass within the mirror pairs $(h, -h)$, but not across pairs. If both $h$ and $-h$ are poor classifiers and their joint prior mass is high, there is no way to remedy this in the posterior.

Lorenzen et al. [2019] have shown that for standard random forests applied to several UCI datasets the first order bound is typically tighter than the various forms of C-bounds proposed by Germain et al. [2015]. However, the first order approach has its own limitations. While it is possible to minimize the bound [Thiemann et al., 2017], it ignores the correlation of errors and minimization of the bound concentrates the weight on a few top classifiers and reduces the power of the ensemble. Our experiments show that minimization of the first order bound typically leads to deterioration of the test error.

We propose a novel analysis of the risk of weighted majority vote in multiclass classification, which addresses the weaknesses of previous methods. The new analysis is based on a second order Markov's inequality, $\mathbb{P}(Z \geq \varepsilon) \leq \mathbb{E}\left[Z^2\right] / \varepsilon^2$, which can be seen as a relaxation of the Chebyshev-Cantelli inequality. We use the inequality to bound the expected loss of weighted majority vote by four times the expected *tandem loss* of the corresponding randomized classifier: The tandem loss measures the probability that two hypotheses drawn independently by the randomized classifier simultaneously err on a sample. Hence, it takes correlation of errors into account. We then use PAC-Bayesian analysis to bound the expected tandem loss in terms of its empirical counterpart and provide a procedure for minimizing the bound and optimizing the weighting. We show that the bound is reasonably tight and that, in contrast to the first order bound, minimization of the bound typically does not deteriorate the performance of the majority vote on new data.

We also present a specialized version of the bound for binary classification, which takes advantage of unlabeled data. It expresses the expected tandem loss in terms of a difference between the expected loss and half the expected disagreement between pairs of hypotheses. In the binary case the disagreements do not depend on the labels and can be estimated from unlabeled data, whereas the loss of a randomized classifier is a first order quantity, which is easier to estimate than the tandem loss. We note, however, that the specialized version only gives advantage over the general one when the amount of unlabeled data is considerably larger than the amount of labeled data.

## 2 General problem setup

**Multiclass classification**    Let $S = \{(X_1, Y_1), \ldots, (X_n, Y_n)\}$ be an independent identically distributed sample from $\mathcal{X} \times \mathcal{Y}$, drawn according to an unknown distribution $D$, where $\mathcal{Y}$ is finite and $\mathcal{X}$ is arbitrary. A hypothesis is a function $h : \mathcal{X} \to \mathcal{Y}$, and $\mathcal{H}$ denotes a space of hypotheses. We evaluate the quality of a hypothesis $h$ by the 0-1 loss $\ell(h(X), Y) = \mathbb{1}(h(X) \neq Y)$, where $\mathbb{1}(\cdot)$ is the indicator function. The expected loss of $h$ is denoted by $L(h) = \mathbb{E}_{(X,Y) \sim D}[\ell(h(X), Y)]$ and the empirical loss of $h$ on a sample $S$ of size $n$ is denoted by $\hat{L}(h, S) = \frac{1}{n} \sum_{i=1}^n \ell(h(X_i), Y_i)$.

**Randomized classifiers**    A *randomized classifier* (a.k.a. Gibbs classifier) associated with a distribution $\rho$ on $\mathcal{H}$, for each input $X$ randomly draws a hypothesis $h \in \mathcal{H}$ according to $\rho$ and predicts $h(X)$. The expected loss of a randomized classifier is given by $\mathbb{E}_{h \sim \rho}[L(h)]$ and the empirical loss by

$\mathbb{E}_{h\sim\rho}[\hat{L}(h,S)]$. To simplify the notation we use $\mathbb{E}_D[\cdot]$ as a shorthand for $\mathbb{E}_{(X,Y)\sim D}[\cdot]$ and $\mathbb{E}_\rho[\cdot]$ as a shorthand for $\mathbb{E}_{h\sim\rho}[\cdot]$.

**Ensemble classifiers and majority vote**   Ensemble classifiers predict by taking a weighted aggregation of predictions by hypotheses from $\mathcal{H}$. The $\rho$-weighted majority vote $\mathrm{MV}_\rho$ predicts $\mathrm{MV}_\rho(X) = \arg\max_{y\in\mathcal{Y}} \mathbb{E}_\rho[\mathbb{1}(h(X) = y)]$, where ties can be resolved arbitrarily.

If majority vote makes an error, we know that at least a $\rho$-weighted half of the classifiers have made an error and, therefore, $\ell(\mathrm{MV}_\rho(X), Y) \leq \mathbb{1}(\mathbb{E}_\rho[\mathbb{1}(h(X) \neq Y)] \geq 0.5)$. This observation leads to the well-known first order oracle bound for the loss of weighted majority vote.

**Theorem 1** (First Order Oracle Bound)**.**

$$L(\mathrm{MV}_\rho) \leq 2\mathbb{E}_\rho[L(h)].$$

*Proof.* We have $L(\mathrm{MV}_\rho) = \mathbb{E}_D[\ell(\mathrm{MV}_\rho(X), Y)] \leq \mathbb{P}(\mathbb{E}_\rho[\mathbb{1}(h(X) \neq Y)] \geq 0.5)$. By applying Markov's inequality to random variable $Z = \mathbb{E}_\rho[\mathbb{1}(h(X) \neq Y)]$ we have:

$$L(\mathrm{MV}_\rho) \leq \mathbb{P}(\mathbb{E}_\rho[\mathbb{1}(h(X) \neq Y)] \geq 0.5) \leq 2\mathbb{E}_D[\mathbb{E}_\rho[\mathbb{1}(h(X) \neq Y)]] = 2\mathbb{E}_\rho[L(h)]. \qquad \square$$

PAC-Bayesian analysis can be used to bound $\mathbb{E}_\rho[L(h)]$ in Theorem 1 in terms of $\mathbb{E}_\rho[\hat{L}(h,S)]$, thus turning the oracle bound into an empirical one. The disadvantage of the first order approach is that $\mathbb{E}_\rho[L(h)]$ ignores correlations of predictions, which is the main power of the majority vote.

# 3   New second order oracle bounds for the majority vote

The key novelty of our approach is using a second order Markov's inequality: for a non-negative random variable $Z$ and $\varepsilon > 0$, we have $\mathbb{P}(Z \geq \varepsilon) = \mathbb{P}(Z^2 \geq \varepsilon^2) \leq \varepsilon^{-2}\mathbb{E}[Z^2]$. We define the *tandem loss* of two hypotheses $h$ and $h'$ on a sample $(X, Y)$ by $\ell(h(X), h'(X), Y) = \mathbb{1}(h(X) \neq Y \wedge h'(X) \neq Y)$. (Lacasse et al. [2007] and Germain et al. [2015] use the term joint error for this quantity.) The tandem loss counts an error on a sample $(X, Y)$ only if both $h$ and $h'$ err on it. The *expected tandem loss* is defined by

$$L(h, h') = \mathbb{E}_D[\mathbb{1}(h(X) \neq Y \wedge h'(X) \neq Y)].$$

The following lemma, given as equation (7) by Lacasse et al. [2007] without a proof, relates the expectation of the second moment of the standard loss to the expected tandem loss. We use $\rho^2$ as a shorthand for the product distribution $\rho \times \rho$ over $\mathcal{H} \times \mathcal{H}$ and the shorthand $\mathbb{E}_{\rho^2}[L(h, h')] = \mathbb{E}_{h\sim\rho, h'\sim\rho}[L(h, h')]$.

**Lemma 2.** *In multiclass classification*

$$\mathbb{E}_D[\mathbb{E}_\rho[\mathbb{1}(h(X) \neq Y)]^2] = \mathbb{E}_{\rho^2}[L(h, h')].$$

A proof is provided in Appendix A. A combination of second order Markov's inequality with Lemma 2 leads to the following result.

**Theorem 3** (Second Order Oracle Bound)**.** *In multiclass classification*

$$L(\mathrm{MV}_\rho) \leq 4\mathbb{E}_{\rho^2}[L(h, h')]. \tag{1}$$

*Proof.* By second order Markov's inequality applied to $Z = \mathbb{E}_\rho[\mathbb{1}(h(X) \neq Y)]$ and Lemma 2:

$$L(\mathrm{MV}_\rho) \leq \mathbb{P}(\mathbb{E}_\rho[\mathbb{1}(h(X) \neq Y)] \geq 0.5) \leq 4\mathbb{E}_D[\mathbb{E}_\rho[\mathbb{1}(h(X) \neq Y)]^2] = 4\mathbb{E}_{\rho^2}[L(h, h')]. \quad \square$$

## 3.1   A specialized bound for binary classification

We provide an alternative form of Theorem 3, which can be used to exploit unlabeled data in binary classification. We denote the *expected disagreement* between hypotheses $h$ and $h'$ by $\mathbb{D}(h, h') = \mathbb{E}_D[\mathbb{1}(h(X) \neq h'(X))]$ and express the tandem loss in terms of standard loss and disagreement. (The lemma is given as equation (8) by Lacasse et al. [2007] without a proof.)

**Lemma 4.** *In binary classification*

$$\mathbb{E}_{\rho^2}[L(h, h')] = \mathbb{E}_\rho[L(h)] - \frac{1}{2}\mathbb{E}_{\rho^2}[\mathbb{D}(h, h')].$$

A proof of the lemma is provided in Appendix A. The lemma leads to the following result.

**Theorem 5** (Second Order Oracle Bound for Binary Classification)**.** *In binary classification*

$$L(\mathrm{MV}_\rho) \leq 4\mathbb{E}_\rho[L(h)] - 2\mathbb{E}_{\rho^2}[\mathbb{D}(h, h')]. \tag{2}$$

*Proof.* The theorem follows by plugging the result of Lemma 4 into Theorem 3. □

The advantage of the alternative way of writing the bound is the possibility of using unlabeled data for estimation of $\mathbb{D}(h, h')$ in binary prediction (see also Germain et al., 2015). We note, however, that estimation of $\mathbb{E}_{\rho^2}[\mathbb{D}(h, h')]$ has a slow convergence rate, as opposed to $\mathbb{E}_{\rho^2}[L(h, h')]$, which has a fast convergence rate. We discuss this point in Section 4.4.

### 3.2 Comparison with the first order oracle bound

From Theorems 1 and 5 we see that in binary classification the second order bound is tighter when $\mathbb{E}_{\rho^2}[\mathbb{D}(h, h')] > \mathbb{E}_\rho[L(h)]$. Below we provide a more detailed comparison of Theorems 1 and 3 in the worst, the best, and the independent cases. The comparison only concerns the oracle bounds, whereas estimation of the oracle quantities, $\mathbb{E}_\rho[L(h)]$ and $\mathbb{E}_{\rho^2}[L(h, h')]$, is discussed in Section 4.4.

**The worst case**   Since $\mathbb{E}_{\rho^2}[L(h, h')] \leq \mathbb{E}_\rho[L(h)]$ the second order bound is at most twice worse than the first order bound. The worst case happens, for example, if all hypotheses in $\mathcal{H}$ give identical predictions. Then $\mathbb{E}_{\rho^2}[L(h, h')] = \mathbb{E}_\rho[L(h)] = L(\mathrm{MV}_\rho)$ for all $\rho$.

**The best case**   Imagine that $\mathcal{H}$ consists of $M \geq 3$ hypotheses, such that each hypothesis errs on $1/M$ of the sample space (according to the distribution $D$) and that the error regions are disjoint. Then $L(h) = 1/M$ for all $h$ and $L(h, h') = 0$ for all $h \neq h'$ and $L(h, h) = 1/M$. For a uniform distribution $\rho$ on $\mathcal{H}$ the first order bound is $2\mathbb{E}_\rho[L(h)] = 2/M$ and the second order bound is $4\mathbb{E}_{\rho^2}[L(h, h')] = 4/M^2$ and $L(\mathrm{MV}_\rho) = 0$. In this case the second order bound is an order of magnitude tighter than the first order.

**The independent case**   Assume that all hypotheses in $\mathcal{H}$ make independent errors and have the same error rate, $L(h) = L(h')$ for all $h$ and $h'$. Then for $h \neq h'$ we have $L(h, h') = \mathbb{E}_D[\mathbb{1}(h(X) \neq Y \wedge h'(X) \neq Y)] = \mathbb{E}_D[\mathbb{1}(h(X) \neq Y)\mathbb{1}(h'(X) \neq Y)] = \mathbb{E}_D[\mathbb{1}(h(X) \neq Y)]\mathbb{E}_D[\mathbb{1}(h'(X) \neq Y)] = L(h)^2$ and $L(h, h) = L(h)$. For a uniform distribution $\rho$ the second order bound is $4\mathbb{E}_{\rho^2}[L(h, h')] = 4(L(h)^2 + \frac{1}{M}L(h)(1 - L(h)))$ and the first order bound is $2\mathbb{E}_\rho[L(h)] = 2L(h)$. Assuming that $M$ is large, so that we can ignore the second term in the second order bound, we obtain that it is tighter for $L(h) < 1/2$ and looser otherwise. The former is the interesting regime, especially in binary classification.

In Appendix B we give additional intuition about Theorems 1 and 3 by providing an alternative derivation.

### 3.3 Comparison with the oracle C-bound

The oracle C-bound is an alternative second order bound based on Chebyshev-Cantelli inequality (Theorem C.13 in the appendix). It was first derived for binary classification by Lacasse et al. [2007, Theorem 2] and several alternative forms were proposed by Germain et al. [2015, Theorem 11]. Laviolette et al. [2017, Corollary 1] extended the result to multiclass classification. To facilitate the comparison with our results we write the bound in terms of the tandem loss. In Appendix D we provide a direct derivation of Theorem 6 from Chebyshev-Cantelli inequality and in Appendix E we show that it is equivalent to prior forms of the oracle C-bound.

**Theorem 6** (C-tandem Oracle Bound)**.** *If $\mathbb{E}_\rho[L(h)] < 1/2$, then*

$$L(\mathrm{MV}_\rho) \leq \frac{\mathbb{E}_{\rho^2}[L(h, h')] - \mathbb{E}_\rho[L(h)]^2}{\mathbb{E}_{\rho^2}[L(h, h')] - \mathbb{E}_\rho[L(h)] + \frac{1}{4}}.$$

The theorem is essentially identical to the first form of oracle C-bound by Lacasse et al. [2007, Theorem 2] and, as we show, it holds for multiclass classification. In Appendix C we show that the second order Markov's inequality behind Theorem 3 is a relaxation of Chebyshev-Cantelli inequality. Therefore, the oracle C-bound is always at least as tight as the second order oracle bound in Theorem 3.

In particular, Germain et al. show that if the classifiers make independent errors and their error rates are identical and below 1/2, the oracle C-bound converges to zero with the growth of the number of classifiers, whereas, as we have shown above, the bound in Theorem 3 only converges to $4L(h)^2$. However, the oracle C-bound has $\mathbb{E}_{\rho^2}[L(h, h')]$ and $\mathbb{E}_{\rho}[L(h)]$ in the denominator, which comes as a significant disadvantage in its estimation from data and minimization [Lorenzen et al., 2019], as we also show in our empirical evaluation.

## 4 Second order PAC-Bayesian bounds for the weighted majority vote

We apply PAC-Bayesian analysis to transform oracle bounds from the previous section into empirical bounds. The results are based on the following two theorems, where we use $\mathrm{KL}(\rho\|\pi)$ to denote the Kullback-Leibler divergence between distributions $\rho$ and $\pi$ and $\mathrm{kl}(p\|q)$ to denote the Kullback-Leibler divergence between two Bernoulli distributions with biases $p$ and $q$.

**Theorem 7** (PAC-Bayes-kl Inequality, Seeger, 2002). *For any probability distribution $\pi$ on $\mathcal{H}$ that is independent of $S$ and any $\delta \in (0, 1)$, with probability at least $1 - \delta$ over a random draw of a sample $S$, for all distributions $\rho$ on $\mathcal{H}$ simultaneously:*

$$\mathrm{kl}\left(\mathbb{E}_{\rho}[\hat{L}(h, S)]\middle\|\mathbb{E}_{\rho}[L(h)]\right) \leq \frac{\mathrm{KL}(\rho\|\pi) + \ln(2\sqrt{n}/\delta)}{n}. \tag{3}$$

The next theorem provides a relaxation of the PAC-Bayes-kl inequality, which is more convenient for optimization. The upper bound is due to Thiemann et al. [2017] and the lower bound follows by an almost identical derivation, see Appendix F. Both results are based on the refined Pinsker's lower bound for the kl-divergence. Since both the upper and the lower bound are deterministic relaxations of PAC-Bayes-kl, they hold simultaneously with no need to take a union bound over the two statements.

**Theorem 8** (PAC-Bayes-$\lambda$ Inequality, Thiemann et al., 2017). *For any probability distribution $\pi$ on $\mathcal{H}$ that is independent of $S$ and any $\delta \in (0, 1)$, with probability at least $1 - \delta$ over a random draw of a sample $S$, for all distributions $\rho$ on $\mathcal{H}$ and all $\lambda \in (0, 2)$ and $\gamma > 0$ simultaneously:*

$$\mathbb{E}_{\rho}[L(h)] \leq \frac{\mathbb{E}_{\rho}[\hat{L}(h, S)]}{1 - \frac{\lambda}{2}} + \frac{\mathrm{KL}(\rho\|\pi) + \ln(2\sqrt{n}/\delta)}{\lambda\left(1 - \frac{\lambda}{2}\right)n}, \tag{4}$$

$$\mathbb{E}_{\rho}[L(h)] \geq \left(1 - \frac{\gamma}{2}\right)\mathbb{E}_{\rho}[\hat{L}(h, S)] - \frac{\mathrm{KL}(\rho\|\pi) + \ln(2\sqrt{n}/\delta)}{\gamma n}. \tag{5}$$

### 4.1 A general bound for multiclass classification

We define the *empirical tandem loss*

$$\hat{L}(h, h', S) = \frac{1}{n}\sum_{i=1}^{n}\mathbb{1}(h(X_i) \neq Y_i \wedge h'(X_i) \neq Y_i)$$

and provide a bound on the expected loss of $\rho$-weighted majority vote in terms of the empirical tandem losses.

**Theorem 9.** *For any probability distribution $\pi$ on $\mathcal{H}$ that is independent of $S$ and any $\delta \in (0, 1)$, with probability at least $1 - \delta$ over a random draw of $S$, for all distributions $\rho$ on $\mathcal{H}$ and all $\lambda \in (0, 2)$ simultaneously:*

$$L(\mathrm{MV}_{\rho}) \leq 4\left(\frac{\mathbb{E}_{\rho^2}[\hat{L}(h, h', S)]}{1 - \lambda/2} + \frac{2\,\mathrm{KL}(\rho\|\pi) + \ln(2\sqrt{n}/\delta)}{\lambda(1 - \lambda/2)n}\right).$$

*Proof.* The theorem follows by using the bound in equation (4) to bound $\mathbb{E}_{\rho^2}[L(h, h')]$ in Theorem 3. We note that $\mathrm{KL}(\rho^2\|\pi^2) = 2\,\mathrm{KL}(\rho\|\pi)$ [Germain et al., 2015, Page 814]. □

It is also possible to use PAC-Bayes-kl to bound $\mathbb{E}_{\rho^2}[L(h, h')]$ in Theorem 3, which actually gives a tighter bound, but the bound in Theorem 9 is more convenient for minimization. Tolstikhin and

Seldin [2013] have shown that for a fixed $\rho$ the expression in Theorem 9 is convex in $\lambda$ and has a closed-form minimizer. In Appendix G we show that for fixed $\lambda$ and $S$ the bound is convex in $\rho$. Although in our applications $S$ is not fixed and the bound is not necessarily convex in $\rho$, a local minimum can still be efficiently achieved by gradient descent. A bound minimization procedure is provided in Appendix H.

## 4.2 A specialized bound for binary classification

We define the *empirical disagreement*

$$\hat{\mathbb{D}}(h, h', S') = \frac{1}{m} \sum_{i=1}^{m} \mathbb{1}(h(X_i) \neq h'(X_i)),$$

where $S' = \{X_1, \ldots, X_m\}$. The set $S'$ may have an overlap with the inputs X of the labeled set $S$, however, $S'$ may include additional unlabeled data. The following theorem bounds the loss of weighted majority vote in terms of empirical disagreements. Due to possibility of using unlabeled data for estimation of disagreements in the binary case, the theorem has the potential of yielding a tighter bound when a considerable amount of unlabeled data is available.

**Theorem 10.** *In binary classification, for any probability distribution $\pi$ on $\mathcal{H}$ that is independent of $S$ and $S'$ and any $\delta \in (0, 1)$, with probability at least $1 - \delta$ over a random draw of $S$ and $S'$, for all distributions $\rho$ on $\mathcal{H}$ and all $\lambda \in (0, 2)$ and $\gamma > 0$ simultaneously:*

$$L(\mathrm{MV}_\rho) \leq 4 \left( \frac{\mathbb{E}_\rho[\hat{L}(h, S)]}{1 - \lambda/2} + \frac{\mathrm{KL}(\rho\|\pi) + \ln(4\sqrt{n}/\delta)}{\lambda(1 - \lambda/2)n} \right)$$
$$- 2 \left( (1 - \gamma/2)\mathbb{E}_{\rho^2}[\hat{\mathbb{D}}(h, h', S')] - \frac{2\,\mathrm{KL}(\rho\|\pi) + \ln(4\sqrt{m}/\delta)}{\gamma m} \right).$$

*Proof.* The theorem follows by using the upper bound in equation (4) to bound $\mathbb{E}_\rho[L(h)]$ and the lower bound in equation (5) to bound $\mathbb{E}_{\rho^2}[\mathbb{D}(h, h')]$ in Theorem 5. We replace $\delta$ by $\delta/2$ in the upper and lower bound and take a union bound over them. □

Using PAC-Bayes-kl to bound $\mathbb{E}_\rho[L(h)]$ and $\mathbb{E}_{\rho^2}[\mathbb{D}(h, h')]$ in Theorem 5 gives a tighter bound, but the bound in Theorem 10 is more convenient for minimisation. The minimization procedure is provided in Appendix H.

## 4.3 Ensemble construction

Thiemann et al. [2017] have proposed an elegant way of constructing finite data-dependent hypothesis spaces that work well with PAC-Bayesian bounds. The idea is to generate multiple splits of a data set $S$ into pairs of subsets $S = T_h \cup S_h$, such that $T_h \cap S_h = \varnothing$. A hypothesis $h$ is then trained on $T_h$ and $\hat{L}(h, S_h)$ provides an unbiased estimate of its loss. The splits cannot depend on the data. Two examples of such splits are splits generated by cross-validation [Thiemann et al., 2017] and splits generated by bagging in random forests, where out-of-bag (OOB) samples provide unbiased estimates of expected losses of individual trees [Lorenzen et al., 2019]. It is possible to train multiple hypotheses with different parameters on each split, as it happens in cross-validation. The resulting set of hypotheses produces an ensemble, and PAC-Bayesian bounds provide generalization bounds for a weighted majority vote of the ensemble and allow optimization of the weighting. There are two minor modifications required: the weighted empirical losses $\mathbb{E}_\rho[\hat{L}(h, S)]$ in the bounds are replaced by weighted validation losses $\mathbb{E}_\rho[\hat{L}(h, S_h)]$, and the sample size $n$ is replaced by the minimal validation set size $n_{\texttt{min}} = \min_h |S_h|$. It is possible to use any data-independent prior, with uniform prior $\pi(h) = 1/|\mathcal{H}|$ being a natural choice in many cases [Thiemann et al., 2017].

For pairs of hypotheses $(h, h')$ we use the overlaps of their validation sets $S_h \cap S_{h'}$ to calculate an unbiased estimate of their tandem loss, $\hat{L}(h, h', S_h \cap S_{h'})$, which replaces $\hat{L}(h, h', S)$ in the bounds. The sample size $n$ is then replaced by $n_{\texttt{min}} = \min_{h,h'}(S_h \cap S_{h'})$.

### 4.4 Comparison of the empirical bounds

We provide a high-level comparison of the empirical first order bound (FO), the new empirical second order bound based on the tandem loss (TND, Theorem 9), and the new empirical second order bound based on disagreements (DIS, Theorem 10). The two key quantities in the comparison are the sample size $n$ in the denominator of the bounds and fast and slow convergence rates for the standard (first order) loss, the tandem loss, and the disagreements. Tolstikhin and Seldin [2013] have shown that if we optimize $\lambda$ for a given $\rho$, the PAC-Bayes-$\lambda$ bound in equation (4) can be written as

$$\mathbb{E}_\rho[L(h)] \leq \mathbb{E}_\rho[\hat{L}(h,S)] + \sqrt{\frac{2\mathbb{E}_\rho[\hat{L}(h,S)]\left(\mathrm{KL}(\rho\|\pi) + \ln(2\sqrt{n}/\delta)\right)}{n}} + \frac{2\left(\mathrm{KL}(\rho\|\pi) + \ln(2\sqrt{n}/\delta)\right)}{n}.$$

This form of the bound, introduced by McAllester [2003], is convenient for explanation of fast and slow rates. If $\mathbb{E}_\rho[\hat{L}(h,S)]$ is large, then the middle term on the right hand side dominates the complexity and the bound decreases at the rate of $1/\sqrt{n}$, which is known as a *slow rate*. If $\mathbb{E}_\rho[\hat{L}(h,S)]$ is small, then the last term dominates and the bound decreases at the rate of $1/n$, which is known as a *fast rate*.

**FO vs. TND**   The advantage of the FO bound is that the validation sets $S_h$ available for estimation of the first order losses $\hat{L}(h,S_h)$ are larger than the validation sets $S_h \cap S_{h'}$ available for estimation of the tandem losses. Therefore, the denominator $n_{\mathtt{min}} = \min_h |S_h|$ in the FO bound is larger than the denominator $n_{\mathtt{min}} = \min_{h,h'} |S_h \cap S_{h'}|$ in the TND bound. The TND disadvantage can be reduced by using data splits with large validation sets $S_h$ and small training sets $T_h$, as long as small training sets do not overly impact the quality of base classifiers $h$. Another advantage of the FO bound is that its complexity term has $\mathrm{KL}(\rho\|\pi)$, whereas the TND bound has $2\,\mathrm{KL}(\rho\|\pi)$. The advantage of the TND bound is that $\mathbb{E}_{\rho^2}[L(h,h')] \leq E_\rho[L(h)]$ and, therefore, the convergence rate of the tandem loss is typically faster than the convergence rate of the first order loss. The interplay of the estimation advantages and disadvantages, combined with the advantages and disadvantages of the underlying oracle bounds discussed in Section 3.2, depends on the data and the hypothesis space.

**TND vs. DIS**   The advantage of the DIS bound relative to the TND bound is that in presence of a large amount of unlabeled data the disagreements $\mathbb{D}(h,h')$ can be tightly estimated (the denominator $m$ is large) and the estimation complexity is governed by the first order term, $\mathbb{E}_\rho[L(h)]$, which is "easy" to estimate, as discussed above. However, the DIS bound has two disadvantages. A minor one is its reliance on estimation of two quantities, $\mathbb{E}_\rho[L(h)]$ and $\mathbb{E}_{\rho^2}[\mathbb{D}(h,h')]$, which requires a union bound, e.g., replacement of $\delta$ by $\delta/2$. A more substantial one is that the disagreement term is desired to be large, and thus has a slow convergence rate. Since slow convergence rate relates to fast convergence rate as $1/\sqrt{n}$ to $1/n$, as a rule of thumb the DIS bound is expected to outperform TND only when the amount of unlabeled data is at least quadratic in the amount of labeled data, $m > n^2$.

## 5 Empirical evaluation

We studied the empirical performance of the bounds using standard random forests [Breiman, 2001] on a subset of data sets from the UCI and LibSVM repositories [Dua and Graff, 2019, Chang and Lin, 2011]. An overview of the data sets is given in Table I.1 in the appendix. The number of points varied from 3000 to 70000 with dimensions $d < 1000$. For each data set we set aside 20% of the data for a test set $S_{\mathrm{test}}$ and used the remaining data, which we call $S$, for ensemble construction and computation of the bounds. Forests with 100 trees were trained until leaves were pure, using the Gini criterion for splitting and considering $\sqrt{d}$ features in each split. We made 50 repetitions of each experiment and report the mean and standard deviation. In all our experiments $\pi$ was uniform and $\delta = 0.05$. We present two experiments: (1) a comparison of tightness of the bounds applied to uniform weighting, and (2) a comparison of weighting optimization the bounds. Additional experiments, where we explored the effect of using splits with increased validation and decreased training subsets, as suggested in Section 4.4, and where we compared the TND and DIS bounds in presence of unlabeled data, are described in Appendix I.

The python source code for replicating the experiments is available at Github[2].

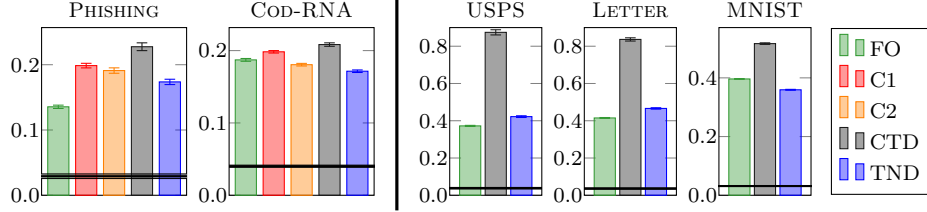

Figure 1: Test risk (black) and the bounds for a uniformly weighted random forest on a subset of binary (left) and multiclass (right) datasets. Plots for the remaining datasets are provided in Figures I.4 and I.5 in the appendix.

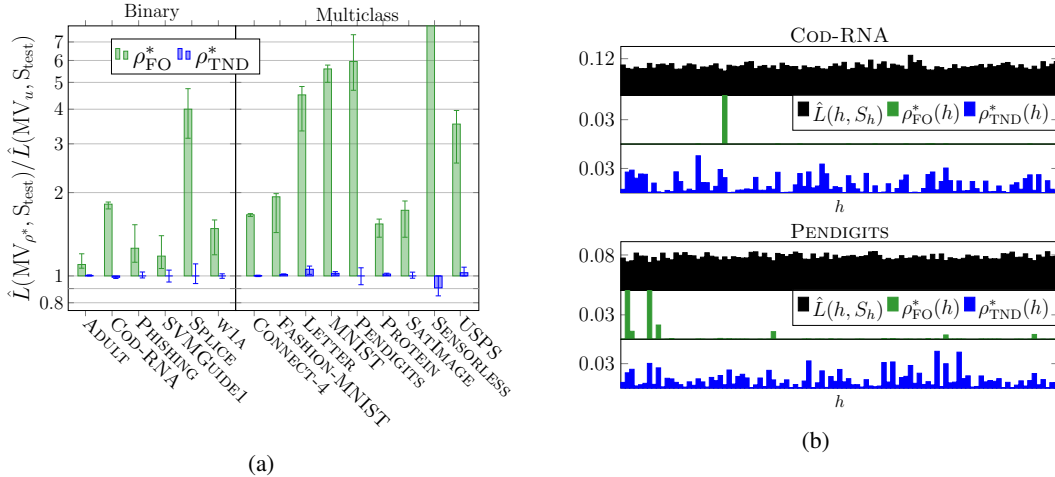

(a)

(b)

Figure 2: (a) The median, 25%, and 75% quantiles of the ratio $\hat{L}(\text{MV}_{\rho^*}, \text{S}_{\text{test}})/\hat{L}(\text{MV}_u, \text{S}_{\text{test}})$ of the test loss of majority vote with optimized weighting $\rho^*$ generated by FO and TND. The plot is on a logarithmic scale. Values above 1 represent degradation in performance on new data and values below 1 represent an improvement. (b) The optimized weights $\rho^*$ generated by FO and TND.

**Uniform weighting** In Figure 1 we compare tightness of FO, C1 and C2 (the two forms of C-bound by Germain et al., 2015, see Appendix E for the oracle forms), the C-tandem bound (CTD, Theorem 6), and TND applied to uniformly weighted random forests on a subset of data sets. The right three plots are multiclass datasets, where C1 and C2 are inapplicable. The outcomes for the remaining datasets are reported in Figures I.4 and I.5 in the appendix. Since no optimization was involved, we used the PAC-Bayes-kl to bound $\mathbb{E}_\rho[L(h)]$, $\mathbb{E}_{\rho^2}[L(h, h')]$, and $\mathbb{E}_{\rho^2}[\mathbb{D}(h, h')]$ in the first and second order bounds, which is tighter than using PAC-Bayes-$\lambda$. The TND bound was the tightest for 5 out of 16 data sets, and provided better guarantees than the C-bounds for 4 out of 7 binary data sets. In most cases, the FO-bound was the tightest.

**Optimization of the weighting** We compared the loss on the test set $\text{S}_{\text{test}}$ and tightness after using the bounds for optimizing the weighting $\rho$. As already discussed, the C-bounds are not suitable for optimization (see also Lorenzen et al., 2019) and, therefore, excluded from the comparison. We used the PAC-Bayes-$\lambda$ form of the bounds for $\mathbb{E}_\rho[L(h)]$, $\mathbb{E}_{\rho^2}[L(h, h')]$, and $\mathbb{E}_{\rho^2}[\mathbb{D}(h, h')]$ for optimization of $\rho$ and then used the PAC-Bayes-kl form of the bounds for computing the final bound with the optimized $\rho$. Optimization details are provided in Appendix H.

Figure 2a compares the ratio of the loss of majority vote with optimized weighting to the loss of majority vote with uniform weighting on $\text{S}_{\text{test}}$ for $\rho^*$ found by minimization of FO and TND. The numerical values are given in Table I.6 in the appendix. While both bounds tighten with optimization, we observed that optimization of FO considerably weakens the performance on $\text{S}_{\text{test}}$ for all datasets, whereas optimization of TND did not have this effect and in some cases even improved the outcome. Figure 2b shows optimized distributions for two sample data sets. It is clearly seen that FO placed

all the weight on a few top trees, while TND hedged the bets on multiple trees. The two figures demonstrate that the new bound correctly handled interactions between voters, as opposed to FO.

## 6 Discussion

We have presented a new analysis of the weighted majority vote, which provides a reasonably tight generalization guarantee and can be used to guide optimization of the weights. The analysis has been applied to random forests, where the bound can be computed using out-of-bag samples with no need for a dedicated hold-out validation set, thus making highly efficient use of the data. We have shown that in contrary to the commonly used first order bound, minimization of the new bound does not lead to deterioration of the test error, confirming that the analysis captures the cancellation of errors, which is the core of the majority vote.

## Acknowledgments and Disclosure of Funding

We thank Omar Rivasplata and the anonymous reviewers for their suggestions for manuscript improvements. AM is funded by the Spanish Ministry of Science, Innovation and Universities under the projects TIN2016-77902-C3-3-P and PID2019-106758GB-C32, and by a Jose Castillejo scholarship CAS19/00279. SSL acknowledges funding by the Danish Ministry of Education and Science, Digital Pilot Hub and Skylab Digital. CI acknowledges support by the Villum Foundation through the project Deep Learning and Remote Sensing for Unlocking Global Ecosystem Resource Dynamics (DeReEco). YS acknowledges support by the Independent Research Fund Denmark, grant number 0135-00259B.

## Broader impact

Ensemble classifiers, in particular random forests, are among the most important tools in machine learning [Fernández-Delgado et al., 2014, Zhu, 2015], which are very frequently applied in practice [e.g., Chen and Guestrin, 2016, Hoch, 2015, Puurula et al., 2014, Stallkamp et al., 2012]. Our study provides generalization guarantees for random forests and a method for tuning the weights of individual trees within a forest, which can lead to even higher accuracies. The result is of high practical relevance.

Given that machine learning models are increasingly used to make decisions that have a strong impact on society, industry, and individuals, it is important that we have a good theoretical understanding of the employed methods and are able to provide rigorous guarantees for their performance. And here lies the strongest contribution of the line of research followed in our study, in which we derive rigorous bounds on the generalization error of random forests and other ensemble methods for multiclass classification.

## Footnotes

[2]`https://github.com/StephanLorenzen/MajorityVoteBounds`

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
