[Supplementary Material]

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

# A  Proof of Lemmas 2 and 4

*Proof of Lemma 2.*

$$\mathbb{E}_D[\mathbb{E}_\rho[\mathbb{1}(h(X) \neq Y)]^2] = \mathbb{E}_D[\mathbb{E}_\rho[\mathbb{1}(h(X) \neq Y)]\mathbb{E}_\rho[\mathbb{1}(h(X) \neq Y)]] \qquad (A.6)$$
$$= \mathbb{E}_D[\mathbb{E}_{\rho^2}[\mathbb{1}(h(X) \neq Y)\mathbb{1}(h'(X) \neq Y)]]$$
$$= \mathbb{E}_D[\mathbb{E}_{\rho^2}[\mathbb{1}(h(X) \neq Y \wedge h'(X) \neq Y)]]$$
$$= \mathbb{E}_{\rho^2}[\mathbb{E}_D[\mathbb{1}(h(X) \neq Y \wedge h'(X) \neq Y)]]$$
$$= \mathbb{E}_{\rho^2}[L(h, h')].$$

$\square$

*Proof of Lemma 4.* Picking from (A.6), we have

$$\mathbb{E}_\rho[\mathbb{1}(h(X) \neq Y)]\mathbb{E}_\rho[\mathbb{1}(h(X) \neq Y)] = \mathbb{E}_\rho[\mathbb{1}(h(X) \neq Y)](1 - \mathbb{E}_\rho[(1 - \mathbb{1}(h(X) \neq Y))]$$
$$= \mathbb{E}_\rho[\mathbb{1}(h(X) \neq Y)] - \mathbb{E}_\rho[\mathbb{1}(h(X) \neq Y)]\mathbb{E}_\rho[\mathbb{1}(h(X) = Y)]$$
$$= \mathbb{E}_\rho[\mathbb{1}(h(X) \neq Y)] - \mathbb{E}_{\rho^2}[\mathbb{1}(h(X) \neq Y \wedge h'(X) = Y)]$$
$$= \mathbb{E}_\rho[\mathbb{1}(h(X) \neq Y)] - \frac{1}{2}\mathbb{E}_{\rho^2}[\mathbb{1}(h(X) \neq h'(X))].$$

By taking expectation with respect to $D$ on both sides and applying Lemma 2 to the left hand side, we obtain:

$$\mathbb{E}_{\rho^2}[L(h, h')] = \mathbb{E}_D[\mathbb{E}_\rho[\mathbb{1}(h(X) \neq Y)] - \frac{1}{2}\mathbb{E}_{\rho^2}[\mathbb{1}(h(X) \neq h'(X))]] = \mathbb{E}_\rho[L(h)] - \frac{1}{2}\mathbb{E}_{\rho^2}[\mathbb{D}(h, h')].$$

$\square$

# B  An alternative derivation of Theorems 1 and 3 using relaxations of the indicator function

Figure B.3: Relaxations of the indicator function.

In this section, we provide an alternative derivation of Theorems 1 and 3 using relaxations of the indicator function. The alternative derivation may provide additional intuition about the method and this is how we initially have arrived to the results.

As explained in Section 2, if majority vote makes an error, then at least a $\rho$-weighted half of the classifiers have made an error. Therefore, we have $\ell(\text{MV}_\rho(X), Y) \leq \mathbb{1}(\mathbb{E}_\rho[\mathbb{1}(h(X) \neq Y)] \geq 0.5)$.

The first order bound can be derived from a first order relaxation of the indicator function. For any $w \in [0,1]$ we have $\mathbb{1}(w \geq 0.5) \leq 2w$, see Figure B.3. Taking $w = \mathbb{E}_\rho[\mathbb{1}(h(X) \neq Y)]$ we have

$$
\begin{aligned}
L(\mathrm{MV}_\rho) &\leq \mathbb{E}_D[\mathbb{1}(\mathbb{E}_\rho[\mathbb{1}(h(X) \neq Y)] \geq 0.5)] \\
&\leq 2\mathbb{E}_D[\mathbb{E}_\rho[\mathbb{1}(h(X) \neq Y)]] = 2\mathbb{E}_\rho[\mathbb{E}_D[\mathbb{1}(h(X) \neq Y)]] = 2\mathbb{E}_\rho[L(h)],
\end{aligned}
$$

which gives the result in Theorem 1.

The second order bound can be derived from a second order relaxation of the indicator function. We use the inequality $\mathbb{1}(w \geq 0.5) \leq 4w^2$, which holds for all $w \in [0,1]$, see Figure B.3. As before, we take $w = \mathbb{E}_\rho[\mathbb{1}(h(X) \neq Y)]$. Then, we have

$$
L(\mathrm{MV}_\rho) \leq \mathbb{E}_D[\mathbb{1}(\mathbb{E}_\rho[\mathbb{1}(h(X) \neq Y)] \geq 0.5)] \leq 4\mathbb{E}_D[\mathbb{E}_\rho[\mathbb{1}(h(X) \neq Y)]^2] = 4\mathbb{E}_{\rho^2}[L(h,h')],
$$

where the last equality is by Lemma 2.

## C  Relation between second order Markov's and Chebyshev-Cantelli inequalities

In this section we show that second order Markov's inequality is a relaxation of Chebyshev-Cantelli inequality. In order to emphasize the relation between the proofs of Theorems 1 and 3 in the body and in the previous section, we provide a direct derivation of Markov's and second order Markov's inequalities using relaxations of the indicator function. For any non-negative random variable $X$ and $\varepsilon > 0$ we have:

$$
\mathbb{1}(X \geq \varepsilon) \leq \frac{1}{\varepsilon}X,
$$

$$
\mathbb{1}(X \geq \varepsilon) \leq \frac{1}{\varepsilon^2}X^2.
$$

We use these inequalities to recover the well-known Markov's inequality and prove the second order Markov's inequality.

**Theorem C.11** (Markov's Inequality). *For a non-negative random variable $X$ and $\varepsilon > 0$*

$$
\mathbb{P}(X \geq \varepsilon) \leq \frac{\mathbb{E}\,[X]}{\varepsilon}.
$$

*Proof.*

$$
\mathbb{P}(X \geq \varepsilon) = \mathbb{E}\,[\mathbb{1}(X \geq \varepsilon)] \leq \frac{\mathbb{E}\,[X]}{\varepsilon}.
$$

$\square$

**Theorem C.12** (Second order Markov's inequality). *For a non-negative random variable $X$ and $\varepsilon > 0$*

$$
\mathbb{P}(X \geq \varepsilon) \leq \frac{\mathbb{E}\,[X^2]}{\varepsilon^2}. \tag{C.7}
$$

*Proof.*

$$
\mathbb{P}(X \geq \varepsilon) = \mathbb{E}\,[\mathbb{1}(X \geq \varepsilon)] \leq \frac{\mathbb{E}\,[X^2]}{\varepsilon^2}.
$$

$\square$

We also cite Chebyshev-Cantelli inequality without a proof. For a proof see, for example, Devroye et al. [1996].

**Theorem C.13** (Chebyshev-Cantelli inequality). *For a real-valued random variable $X$ and $\varepsilon > 0$*

$$
\mathbb{P}(X - \mathbb{E}\,[X] \geq \varepsilon) \leq \frac{\mathbb{V}[X]}{\varepsilon^2 + \mathbb{V}[X]}, \tag{C.8}
$$

*where $\mathbb{V}[X] = \mathbb{E}\,[X^2] - \mathbb{E}\,[X]^2$ is the variance of $X$.*

Finally, we show that second order Markov's inequality is a relaxation of Chebyshev-Cantelli inequality.

**Lemma C.14.** *The second-order Markov's inequality* (C.7) *is a relaxation of Chebyshev-Cantelli inequality* (C.8).

*Proof.* We show that inequality (C.8) is always at least as tight as inequality (C.7). The inequality (C.7) is only non-trivial when $\mathbb{E}\left[X\right] < \varepsilon$, so for the comparison we can assume that $\mathbb{E}\left[X\right] < \varepsilon$. By (C.8) we then have:

$$\mathbb{P}(X \geq \varepsilon) = \mathbb{P}(X - \mathbb{E}\left[X\right] \geq \varepsilon - \mathbb{E}\left[X\right]) \leq \frac{\mathbb{V}[X]}{(\varepsilon - \mathbb{E}\left[X\right])^2 + \mathbb{V}[X]} = \frac{\mathbb{E}\left[X^2\right] - \mathbb{E}\left[X\right]^2}{\varepsilon^2 - 2\varepsilon\mathbb{E}\left[X\right] + \mathbb{E}\left[X^2\right]}$$

Thus, we need to compare

$$\frac{\mathbb{E}\left[X^2\right] - \mathbb{E}\left[X\right]^2}{\varepsilon^2 - 2\varepsilon\mathbb{E}\left[X\right] + \mathbb{E}\left[X^2\right]} \quad \text{vs.} \quad \frac{\mathbb{E}\left[X^2\right]}{\varepsilon^2}.$$

This is equivalent to the following row of comparisons:

$$
\begin{aligned}
(\mathbb{E}\left[X^2\right] - \mathbb{E}\left[X\right]^2)\varepsilon^2 \quad &\text{vs.} \quad \mathbb{E}\left[X^2\right](\varepsilon^2 - 2\varepsilon\mathbb{E}\left[X\right] + \mathbb{E}\left[X^2\right]) \\
-\mathbb{E}\left[X\right]^2\varepsilon^2 \quad &\text{vs.} \quad \mathbb{E}\left[X^2\right](-2\varepsilon\mathbb{E}\left[X\right] + \mathbb{E}\left[X^2\right]) \\
0 \quad &\text{vs.} \quad \mathbb{E}\left[X\right]^2\varepsilon^2 - 2\varepsilon\mathbb{E}\left[X\right]\mathbb{E}\left[X^2\right] + \mathbb{E}\left[X^2\right]^2 \\
0 \quad &\leq \quad (\mathbb{E}\left[X\right]\varepsilon - \mathbb{E}\left[X^2\right])^2,
\end{aligned}
$$

which completes the proof. $\square$

# D  A proof of Theorem 6

We provide a direct proof of Theorem 6 using Chebyshev-Cantelli inequality.

*Proof.* We apply Chebyshev-Cantelli inequality to $\mathbb{E}_\rho[\mathbb{1}(h(X) \neq Y)]$:

$$
\begin{aligned}
L(\mathrm{MV}_\rho) &\leq \mathbb{P}\left(\mathbb{E}_\rho[\mathbb{1}(h(X) \neq Y)] \geq \frac{1}{2}\right) \\
&= \mathbb{P}\left(\mathbb{E}_\rho[\mathbb{1}(h(X) \neq Y)] - \mathbb{E}_\rho[L(h)] \geq \frac{1}{2} - \mathbb{E}_\rho[L(h)]\right) \\
&\leq \frac{\mathbb{E}_{\rho^2}[L(h,h')] - \mathbb{E}_\rho[L(h)]^2}{\left(\frac{1}{2} - \mathbb{E}_\rho[L(h)]\right)^2 + \mathbb{E}_{\rho^2}[L(h,h')] - \mathbb{E}_\rho[L(h)]^2} \\
&= \frac{\mathbb{E}_{\rho^2}[L(h,h')] - \mathbb{E}_\rho[L(h)]^2}{\mathbb{E}_{\rho^2}[L(h,h')] - \mathbb{E}_\rho[L(h)] + \frac{1}{4}}.
\end{aligned}
$$

$\square$

# E  Equivalence of Theorem 6 to prior forms of the oracle C-bound

In this section we show that the C-tandem oracle bound in Theorem 6 is equivalent to prior forms of the oracle C-bound.

## E.1  Equivalence to Corollary 1 of Laviolette et al. [2017]

Laviolette et al. write their oracle C-bound in terms of an $\omega$-margin, denoted by $M_{\rho,\omega}(\mathrm{X},\mathrm{Y})$, which is defined as $M_{\rho,\omega}(X,Y) = \mathbb{E}_\rho[\mathbb{1}(h(X) = Y)] - \omega$, or, equivalently, $M_{\rho,\omega}(X,Y) = (1 - \omega) - \mathbb{E}_\rho[\mathbb{1}(h(X) \neq Y)]$. By simple algebraic manipulations we have the following identities, which show the equivalence to Theorem 6:

$$\underbrace{\frac{\mathbb{E}_{\rho^2}[L(h,h')] - \mathbb{E}_\rho[L(h)]^2}{\mathbb{E}_{\rho^2}[L(h,h')] - \mathbb{E}_\rho[L(h)] + \frac{1}{4}}}_{\text{C-tandem oracle}} = 1 - \frac{\frac{1}{4} - \mathbb{E}_\rho[L(h)] + \mathbb{E}_\rho[L(h)]^2}{\frac{1}{4} - \mathbb{E}_\rho[L(h)] + \mathbb{E}_{\rho^2}[L(h,h')]} = \underbrace{1 - \frac{\left(\mathbb{E}_D[M_{\rho,\frac{1}{2}}(X,Y)]\right)^2}{\mathbb{E}_D[(M_{\rho,\frac{1}{2}}(X,Y))^2]}}_{\text{Oracle C-bound of Laviolette et al.}}.$$

## E.2 Equivalence to Theorem 11 of Germain et al. [2015] in binary classification

In binary classification we can apply Lemma 4 and simple algebraic manipulations to obtain the following identities, which demonstrate equivalence of Theorem 6 and Theorem 11 of Germain et al.:

$$\underbrace{\frac{\mathbb{E}_{\rho^2}[L(h,h')] - \mathbb{E}_\rho[L(h)]^2}{\mathbb{E}_{\rho^2}[L(h,h')] - \mathbb{E}_\rho[L(h)] + \frac{1}{4}}}_{\text{C-tandem oracle}} = \frac{4\mathbb{E}_\rho[L(h)] - 4(\mathbb{E}_\rho[L(h)])^2 - 2\mathbb{E}_{\rho^2}[\mathbb{D}(h,h')]}{1 - 2\mathbb{E}_{\rho^2}[\mathbb{D}(h,h')]}$$

$$= \underbrace{1 - \frac{\left(1 - 2\mathbb{E}_\rho[L(h)]\right)^2}{1 - 2\mathbb{E}_{\rho^2}[\mathbb{D}(h,h')]}}_{\text{C1 oracle}}$$

$$= \underbrace{1 - \frac{\left(1 - (2\mathbb{E}_{\rho^2}[L(h,h')] + \mathbb{E}_{\rho^2}[\mathbb{D}(h,h')])\right)^2}{1 - 2\mathbb{E}_{\rho^2}[\mathbb{D}(h,h')]}}_{\text{C2 oracle}}.$$

The second line is the oracle form of C1 bound of Germain et al. and the last line is the oracle form of their C2 bound.

We note that while all forms of the oracle C-bound are equivalent, their translation into empirical bounds might have different tightness due to varying difficulty of estimation of the oracle quantities $\mathbb{E}_\rho[L(h)]$, $\mathbb{E}_{\rho^2}[L(h,h')]$, and $\mathbb{E}_{\rho^2}[\mathbb{D}(h,h')]$, as discussed in Section 4.4.

# F  A proof of Theorem 8

We provide a proof of the lower bound (5) in Theorem 8. The upper bound (4) has been shown by Thiemann et al. [2017]. The proof of the lower bound follows the same steps as the proof of the upper bound.

*Proof.* We use the following version of refined Pinsker's inequality [Marton, 1996, 1997, Samson, 2000, Boucheron et al., 2013, Lemma 8.4]: for $p > q$

$$\mathrm{kl}(p\|q) \geq (p-q)^2/(2p). \tag{F.9}$$

By application of inequality (F.9), inequality (3) can be relaxed to

$$\mathbb{E}_\rho\left[\hat{L}(h,S)\right] - \mathbb{E}_\rho[L(h)] \leq \sqrt{2\mathbb{E}_\rho\left[\hat{L}(h,S)\right]\frac{\mathrm{KL}(\rho\|\pi) + \ln\frac{2\sqrt{n}}{\delta}}{n}}. \tag{F.10}$$

By using the inequality $\sqrt{xy} \leq \frac{1}{2}\left(\gamma x + \frac{y}{\gamma}\right)$ for all $\gamma > 0$, we have that with probability at least $1 - \delta$ for all $\rho$ and $\gamma > 0$

$$\mathbb{E}_\rho\left[\hat{L}(h,S)\right] - \mathbb{E}_\rho[L(h)] \leq \frac{\gamma}{2}\mathbb{E}_\rho\left[\hat{L}(h,S)\right] + \frac{\mathrm{KL}(\rho\|\pi) + \ln\frac{2\sqrt{n}}{\delta}}{\gamma n}. \tag{F.11}$$

By changing sides

$$\mathbb{E}_\rho[L(h)] \geq \left(1 - \frac{\gamma}{2}\right)\mathbb{E}_\rho\left[\hat{L}(h,S)\right] - \frac{\mathrm{KL}(\rho\|\pi) + \ln\frac{2\sqrt{n}}{\delta}}{\gamma n}.$$

$\square$

# G Positive semi-definiteness of the matrix of empirical tandem losses

In Lemma G.15 below we show that if the empirical tandem losses are evaluated on the same set $S$, then the matrix of empirical tandem losses $\hat{L}_{\text{tnd}}$ with entries $(\hat{L}_{\text{tnd}})_{h,h'} = \hat{L}(h, h', S)$ is positive semi-definite. This implies that for a fixed $\lambda$ the bound in Theorem 9 is convex in $\rho$, because in this case $\mathbb{E}_\rho^2[\hat{L}(h, h', S)] = \rho^T \hat{L}_{\text{tnd}} \rho$ is convex in $\rho$ and $\text{KL}(\rho\|\pi)$ is always convex in $\rho$. (We note, however, that the bound is not necessarily jointly convex in $\rho$ and $\lambda$ and, therefore, alternating minimization of the bound may still converge to a local minimum. While Thiemann et al. [2017] derive conditions under which the PAC-Bayes-$\lambda$ bound for the first order loss is quasiconvex, such analysis of the bound for the second order loss would be more complicated.) In Section G.1 we then provide an example showing that if the tandem losses are evaluated on different sets, as it happens in our case, where the entries are $(\hat{L}_{\text{tnd}})_{h,h'} = \hat{L}(h, h', S_h \cap S_{h'})$, then the matrix of tandem losses is not necessarily positive semi-definite. Therefore, in our case minimization of the bound is only expected to converge to a local minimum.

**Lemma G.15.** *Given $M$ hypotheses and data $S = \{(X_1, Y_1), \ldots, (X_n, Y_n)\}$, the $M \times M$ matrix of empirical tandem losses $\hat{L}_{tnd}$ with entries $(\hat{L}_{tnd})_{h,h'} = \hat{L}(h, h', S)$ is positive semi-definite.*

*Proof.* Define a vector of empirical losses by hypotheses in $\mathcal{H}$ on a sample $(X_i, Y_i)$ by

$$\hat{\ell}_i = \begin{pmatrix} \mathbb{1}(h_1(X_i) \neq Y_i) \\ \vdots \\ \mathbb{1}(h_M(X_i) \neq Y_i) \end{pmatrix}.$$

Then the $(h, h')$ entry of the matrix $\hat{\ell}_i \hat{\ell}_i^T$ is $(\hat{\ell}_i \hat{\ell}_i^T)_{h,h'} = \mathbb{1}(h(X_i) \neq Y_i)\mathbb{1}(h'(X_i) \neq Y_i)$. Thus, the matrix of empirical tandem losses can be written as a mean of outer products

$$\hat{L}_{\text{tnd}} = \frac{1}{n} \sum_{i=1}^n \hat{\ell}_i \hat{\ell}_i^T$$

and is, therefore, positive semi-definite. $\qquad\square$

## G.1 Non positive semi-definite example

If the empirical tandem losses are estimated on different subsets of the data rather than a common set $S$, as in the case of out-of-bag samples, where we take $\hat{L}(h, h', S_h \cap S_{h'})$, the resulting matrix of empirical tandem losses is not necessarily positive semi-definite. Consider the following example with 2 points, 3 hypotheses, and the following losses:

|       | $X_1$ | $X_2$ |
|-------|-------|-------|
| $h_1$ | 1     | 0     |
| $h_2$ | 0     | 1     |
| $h_3$ | 0     | 1     |

If we compute the tandem loss for $h_1$ and $h_2$ on the first point and the tandem loss for $h_1$ and $h_3$ and for $h_2$ and $h_3$ on the second point, and the tandem losses of hypotheses with themselves on all the points, then we have

$$\hat{L}(h, h') = \begin{pmatrix} 0.5 & 0 & 0 \\ 0 & 0.5 & 1 \\ 0 & 1 & 0.5 \end{pmatrix}.$$

This matrix is not positive semi-definite, it has eigenvalues $-0.5, 0.5$, and $1.5$.

# H Gradient-based minimization of the bounds

This section gives details on the optimization of the bounds in Theorems 9 and 10. First, we consider the bound in Theorem 9 and provide a closed form solution for the parameter $\lambda$ given $\rho$ as well as the gradient of the bound w.r.t. $\rho$ for fixed $\lambda$. Then we give the closed form solutions for the parameters $\lambda$ and $\gamma$ given $\rho$ and the gradient w.r.t. $\rho$ for fixed $\lambda$ and $\gamma$ for the bound in Theorem 10. After that, we describe the alternating minimization procedure we applied for optimization in our experiments.

### H.1 Minimization of the bound in Theorem 9

**Optimal $\lambda$ given $\rho$** Given $\rho$, the optimal $\lambda$ in Theorem 9 can be computed following Tolstikhin and Seldin [2013] and Thiemann et al. [2017], because the optimization problem is the same:

$$\lambda = \frac{2}{\sqrt{\frac{2n\mathbb{E}_{\rho^2}[\hat{L}(h,h',S)]}{2\,\mathrm{KL}(\rho\|\pi)+\ln\frac{2\sqrt{n}}{\delta}}+1}+1}.$$

**Gradient with respect to $\rho$ given $\lambda$** Next we calculate the gradient for minimizing the bound in Theorem 9 with respect to $\rho$ under fixed $\lambda$. The minimization is equivalent to minimizing $f(\rho) = \mathbb{E}_{\rho^2}[\hat{L}(h,h',S)] + \frac{2}{\lambda n}\,\mathrm{KL}(\rho\|\pi)$ under the constraint that $\rho$ is a probability distribution. Let $(\nabla f)_h$ for $h \in \mathcal{H}$ denote the component of the gradient corresponding to hypothesis $h$. We also use $\hat{L}_{\mathrm{tnd}}$ to denote the matrix of empirical tandem losses and $\ln\frac{\rho}{\pi}$ to denote the vector with entry corresponding to hypothesis $h$ being $\ln\frac{\rho(h)}{\pi(h)}$. We have:

$$(\nabla f)_h = 2\sum_{h'}\rho(h')\hat{L}(h,h',S) + \frac{2}{\lambda n}\left(1 + \ln\frac{\rho(h)}{\pi(h)}\right),$$

$$\nabla f = 2\left(\hat{L}_{\mathrm{tnd}}\rho + \frac{1}{\lambda n}\left(1 + \ln\frac{\rho}{\pi}\right)\right).$$

### H.2 Minimization of the bound in Theorem 10

**Optimal $\lambda$ and $\gamma$ given $\rho$** The optimal $\lambda$ can be computed as above, because the optimization problem is the same. The only difference is that we have $\delta/2$ instead of $\delta$:

$$\lambda = \frac{2}{\sqrt{\frac{2n\mathbb{E}_{\rho}[\hat{L}(h,S)]}{\mathrm{KL}(\rho\|\pi)+\ln\frac{4\sqrt{n}}{\delta}}+1}+1}.$$

Minimization of the bound in Theorem 10 with fixed $\rho$ with respect to $\gamma$ is equivalent to minimizing $\frac{\gamma}{2}a + \frac{b}{\gamma}$ with $a = \mathbb{E}_{\rho^2}[\hat{\mathbb{D}}(h,h',S')]$ and $b = \frac{2\,\mathrm{KL}(\rho\|\pi)+\ln(4\sqrt{m}/\delta)}{m}$. The minimum is achieved by $\gamma = \sqrt{\frac{2b}{a}}$:

$$\gamma = \sqrt{\frac{4\,\mathrm{KL}(\rho\|\pi) + \ln(16m/\delta^2)}{m\mathbb{E}_{\rho^2}[\hat{D}(h,h',S')]}}.$$

**Gradient with respect to $\rho$** Minimization of the bound with respect to $\rho$ for fixed $\lambda$ and $\gamma$ is equivalent to constrained minimization of $f(\rho) = 2a\mathbb{E}_{\rho}[\hat{L}(h,S)] - b\mathbb{E}_{\rho^2}[\hat{\mathbb{D}}(h,h',S')] + 2c\,\mathrm{KL}(\rho\|\pi)$, where $a = \frac{1}{1-\lambda/2}$, $b = 1 - \gamma/2$, and $c = \frac{1}{\lambda(1-\lambda/2)n} + \frac{1}{\gamma m}$, and the constraint is that $\rho$ is a probability distribution. We use $\hat{L}$ to denote the vector of empirical losses of $h \in \mathcal{H}$ and $\hat{\mathbb{D}}$ to denote the matrix of empirical disagreements. We have:

$$(\nabla f)_h = 2a\hat{L}(h,S) - 2b\sum_{h'}\rho(h')\mathbb{D}(h,h',S) + 2c\left(1 + \ln\frac{\rho}{\pi}\right),$$

$$\nabla f = 2\left(a\hat{L} - b\hat{\mathbb{D}}\rho + c\left(1 + \ln\frac{\rho}{\pi}\right)\right).$$

### H.3 Alternating optimization procedure

In our experiments, we applied an alternating optimization procedure to improve the weighting $\rho$ of the ensemble members as well as the parameters $\lambda$ and, when considering the disagreement, $\gamma$.

Let $M = |\mathcal{H}|$ denote the number of ensemble members. We parameterize $\rho$ by $\tilde{\rho} \in \mathbb{R}^M$ with $\rho = \mathrm{softmax}(\tilde{\rho})$, where $\rho_i = \frac{\exp\tilde{\rho}_i}{\sum_{j=1}^{M}\exp\tilde{\rho}_j}$ for $i = 1,\ldots,M$. This ensures that $\rho$ is a proper probability distribution and allows us to apply unconstrained optimization in the adaption of $\rho$.

Because we are using uniform priors $\pi$ and due to the regularization in terms of the Kullback-Leibler divergence between $\rho$ and $\pi$ in the bounds, excluding $\rho_i \in \{0, 1\}$ for each $i = 1, \ldots, B$ is not a limitation.

Starting from uniform $\rho$ and the corresponding optimal $\lambda$ and, if applicable, $\gamma$, we looped through the following steps: We applied iterative gradient-based optimization of $\rho$ parameterized by $\tilde{\rho}$ until the bound did not improve for 10 iterations. Then we computed the optimal $\lambda$ and, in the case of the DIS bound, $\gamma$ for the optimized $\rho$. We stopped if the change in the bound was smaller than $10^{-9}$. We applied iRProp$^+$ for the gradient based optimization, a first order method with adaptive individual step sizes [Igel and Hüsken, 2003, Florescu and Igel, 2018].

# I  Experiments

This section provides details on the data sets used in the experiments and provides details, additional figures, and numerical values for the empirical evaluations: empirical evaluation of the bounds using a standard random forest with uniform weighting (Section I.2, expanding the first experiment and Figure 1 in the body), and optimization of the weighting of the trees (Section I.3, expanding the second experiment and Figure 2 in the body). We also include additional experiments with *reduced bagging*, where we use less data for construction of each tree in order to leave larger out-of-bag sets for improved estimation of the second order quantities. The diagram below provides an overview of the experiments with references to the relevant subsections.

**A** Comparison of uniformly weighted random forests and random forests with optimized weighting in the full bagging setting: Section I.3, expanding on the experiments in the body of the paper.

**B** Comparison of uniformly weighted random forests with standard (full) and reduced bagging: Section I.4.

**C** Comparison of random forests with optimized weighting in the full and reduced bagging settings: Section I.4

For each experiment, we report the mean and standard deviations of 50 runs. We used standard random forests trained on $S$ (80% of the data) and evaluated on test set $S_{\text{test}}$ (20%). 100 trees were used for each data set, and $\sqrt{d}$ features were considered in each split. The bounds were evaluated on the OOB data, with uniform $\pi$ and $\delta = 0.05$.

Furthermore, Section I.5 presents an empirical evaluation of the DIS bound in the setting with only a small amount of labeled data available and large amounts of unlabeled data. For this experiment, we reserved part of $S$ as unlabeled data and evaluated FO, TND and DIS. We varied the split between labeled training data and unlabeled data and report the means and standard deviations of 20 runs for each split.

Table I.1: Data set overview. $c_{\min}$ and $c_{\max}$ denote the minimum and maximum class frequency.

| Dataset | $N$ | $d$ | $c$ | $c_{\min}$ | $c_{\max}$ | Source |
|---|---|---|---|---|---|---|
| ADULT | 32561 | 123 | 2 | 0.2408 | 0.7592 | LIBSVM (a1a) |
| COD-RNA | 59535 | 8 | 2 | 0.3333 | 0.6667 | LIBSVM |
| CONNECT-4 | 67557 | 126 | 3 | 0.0955 | 0.6583 | LIBSVM |
| FASHION-MNIST | 70000 | 784 | 10 | 0.1000 | 0.1000 | Zalando Research |
| LETTER | 20000 | 16 | 26 | 0.0367 | 0.0406 | UCI |
| MNIST | 70000 | 780 | 10 | 0.0902 | 0.1125 | LIBSVM |
| MUSHROOM | 8124 | 22 | 2 | 0.4820 | 0.5180 | LIBSVM |
| PENDIGITS | 10992 | 16 | 10 | 0.0960 | 0.1041 | LIBSVM |
| PHISHING | 11055 | 68 | 2 | 0.4431 | 0.5569 | LIBSVM |
| PROTEIN | 24387 | 357 | 3 | 0.2153 | 0.4638 | LIBSVM |
| SVMGUIDE1 | 3089 | 4 | 2 | 0.3525 | 0.6475 | LIBSVM |
| SATIMAGE | 6435 | 36 | 6 | 0.0973 | 0.2382 | LIBSVM |
| SENSORLESS | 58509 | 48 | 11 | 0.0909 | 0.0909 | LIBSVM |
| SHUTTLE | 58000 | 9 | 7 | 0.0002 | 0.7860 | LIBSVM |
| SPLICE | 3175 | 60 | 2 | 0.4809 | 0.5191 | LIBSVM |
| USPS | 9298 | 256 | 10 | 0.0761 | 0.1670 | LIBSVM |
| W1A | 49749 | 300 | 2 | 0.0297 | 0.9703 | LIBSVM |

## I.1 Data sets

As mentioned, we considered data sets from the UCI and LibSVM repositories [Dua and Graff, 2019, Chang and Lin, 2011], as well as FASHION-MNIST from Zalando Research[3]. We used data sets with size $3000 \leq N \leq 70000$ and dimension $d \leq 1000$. These relatively large data sets were chosen in order to provide meaningful bounds in the standard bagging setting, where individual trees are trained on $n = 0.8N$ randomly subsampled points with replacement and the size of the overlap of out-of-bag sets is roughly $n/9$. An overview of the data sets is given in Table I.1.

For all experiments, we removed patterns with missing entries and made a stratified split of the data set. For data sets with a training and a test set (SVMGUIDE1, SPLICE, ADULT, W1A, MNIST, SHUTTLE, PENDIGITS, PROTEIN, SATIMAGE, USPS) we combined the training and test sets and shuffled the entire set before splitting.

## I.2 Standard uniformly weighted random forests

This section provides additional figures and numerical values of the bounds computed for the standard uniformly weighted random forest using bagging (Figure 1 in the body), as well as additional statistics for the experiments.

Figures I.4 and I.5 plot the bounds obtained by the standard random forest for the binary and multiclass data sets respectively. Table I.2 reports the means and standard deviations for all data sets. Additional information (randomized loss, tandem loss, etc.) is reported in Table I.3.

TND is tightest for 2 out of 7 binary data sets and 3 out of 10 multiclass data sets, while FO is tightest for the rest. Figure I.6 plots the ratio between the empirical disagreement $\mathbb{E}_{\rho^2}[\hat{\mathbb{D}}(h, h', S_h \cap S_{h'})]$ and the empirical randomized loss $\mathbb{E}_{\rho}[\hat{L}(h, S_h)]$ versus the ratio between the TND and FO bounds. This figure shows that TND bound tends to be tighter than FO when the disagreement is large in relation to the randomized loss. Since the amounts of data $|S_h \cap S_{h'}|$ available for estimation of the tandem losses are considerably smaller than the amounts of data $|S_h|$ available for estimation of the first order losses, the empirical disagreement has to be considerably larger than the empirical loss for TND to take the advantage over FO. This is in agreement with the discussion provided in Sections 3.2 and 4.4.

Comparing TND to the other second order bounds, we see that TND is tighter (or almost as tight) in all cases, except for MUSHROOM, where C1 is tighter. This is due to C1 being given in terms of an upper bound on $\mathbb{E}_{\rho}[L(h)]$ and a lower bound on $\mathbb{E}_{\rho^2}[\mathbb{D}(h, h')]$. With the lower bound being

Figure I.4: Plot of the bounds for binary data sets with the standard uniformly weighted random forests. The test losses are depicted by black lines.

Figure I.5: Plot of the bounds for multiclass data sets with standard uniformly weighted random forests. The test losses are depicted by black lines.

almost zero, we have $C1 \approx 2\,FO$ and since the disagreement is very low, $TND \approx 4\,FO$. We note that even though C1 is tighter than TND in this case, it is still much weaker than FO, because, as it has been discussed in Section 3.2, problems with low disagreement are not well-suited for second order bounds.

## I.3 Standard random forests with optimized weights

This section contains numerical values and additional figures for the optimization experiments provided in the second experiment in the body (Figure 2). FO was optimized using Theorem 8 and the alternating update rules of [Thiemann et al., 2017]. For optimizing TND, we used iRProp[+] [Igel and Hüsken, 2003], see Appendix H. We denote the weights after optimization of FO and TND by $\rho^*_{FO}$ and $\rho^*_{TND}$, respectively.

Figure I.6: Ratio between the empirical disagreement $\mathbb{E}_{\rho^2}[\hat{\mathbb{D}}(h, h', S_h \cap S_{h'})]$ and the empirical randomized loss $\mathbb{E}_{\rho}[\hat{L}(h, S_h)]$ versus the ratio between the TND bound and the FO bound. The data sets Mushroom, Shuttle and Protein are excluded. The first two because the randomized loss is extremely small. And the third one because the bounds are higher than 1.

Figures I.7 and I.8 show the bounds before and after optimization for binary and multiclass data sets respectively. The FO bound achieves higher reduction after minimization, however, as illustrated in both figures and Figure 2 in the body, this improvement comes at the cost of considerable increase of the test loss $L(MV_{\rho_{FO}^*}, S_{test})$. The latter happens because FO places most of the posterior mass on a few top classifiers and diminishes the power of the ensemble, see Figure 2b. The improvement of the TND after minimization is more modest, but on a highly positive side it does not degrade the classifier.

Figure I.7: Comparison of the bounds before (not dotted bars) and after (dotted bars) optimization for the binary data sets. The test risk is shown in black.

Figure I.8: Comparison of the bounds before (not dotted bars) and after (dotted bars) optimization for the multiclass data sets. The test risk is shown in black.

Table I.6 shows the numerical values used in Figure 2a.

## I.4 Random forests with reduced bagging vs. full bagging with uniform and optimized weights

The TND bound depends on the size of overlaps $S_h \cap S_{h'}$, which are used to estimate the tandem losses and define the denominator of the bound. In order to ensure that the overlaps $S_h \cap S_{h'}$ are not too small, it might be beneficial to generate splits with $|S_h|$ of at least $(2/3)n$, so that $|S_h \cap S_{h'}|$ is at least $n/3$. In our application to random forests we reduce the number of sampled points in bagging from $n$ to $n/2$, which increases the number of out-of-bag samples $|S_h|$ from roughly $n/3$ to roughly $(2/3)n$ and the overlaps from roughly $n/9$ to $n/3$. We show that the corresponding decrease in $|T_h|$ leads to a relatively small decrease of prediction quality of individual trees and improves the bounds.

We call the bagging procedure that samples $n$ points with replacement a *standard bagging* or *full bagging* and the procedure that samples $n/2$ points *reduced bagging*. This section presents results for random forests trained with *reduced bagging*, including comparisons to the full bagging setting.

Figure I.13 compares the test risk in the full bagging and the reduced bagging settings with uniform and optimized weights. In both uniform and optimized weights we see a limited increase (and in a few cases even a small decrease) in test risk when reducing the amount of data sampled in bagging, indicating that reduced bagging has relatively minor impact on the quality of a uniformly weighted ensemble. At the same time, Figures I.14, I.9, I.10, I.11, and I.12 show that the bounds are improved in most cases, sometimes considerably.

Table I.4 reports the means and standard deviations for all data sets. Additional information (randomized loss, tandem loss, etc.) is reported in Table I.5. Table I.7 reports the performance of the final majority vote with and without optimized weights.

Figure I.9: Comparison of the bounds in the full (not dotted) and reduced bagging (dotted) setting with uniform weighting for binary data sets. The test risk is shown in black.

## I.5 DIS **bound vs.** TND **bound in presence of unlabeled data**

In this section we compare the tightness of the TND and DIS bounds in a setting, where a lot of unlabeled data is available.

We considered the largest binary data sets ($N > 8000$) from Table I.1. As in the previous setting, 20% of the data, $S_{\text{test}}$, was reserved for testing. The remaining 80%, were split with a fraction $r \in [0, 1]$ of patterns $S$ used for training, and a fraction $(1 - r)$ set aside as unlabeled patterns, $S_u$. Forests with 100 trees were trained with bagging, using the Gini criterion for splitting and considering $\sqrt{d}$ features in each split. We considered values of $r \in \{0.05, 0.1, ..., 0.5\}$. For each split, we repeated the experiment 20 times.

Figure I.15 plots the test risk and FO, TND and DIS bounds as a function of $r$. For each data set, the mean and standard deviation over 20 runs are plotted. In agreement with the discussion in Section 4.4, DIS had the highest advantage over TND when the amount of unlabeled data relative to labeled data was the largest. As the amount of unlabeled data relative to labeled data was decreasing the difference between the bounds became smaller, with TND eventually overtaking DIS in most cases.

Figure I.10: Comparison of the bounds in the full (not dotted) and reduced bagging (dotted) setting with uniform weighting for multiclass data sets. The test risk is shown in black.

Figure I.11: Comparison of the bounds computed for the random forest with optimized weights in the standard bagging (not dotted) and the reduced bagging (dotted) setting on binary data sets.

Figure I.12: Comparison of the bounds computed for the random forest with optimized weights in the standard bagging (not dotted) and the reduced bagging (dotted) setting on multiclass data sets.

Figure I.13: Median, 25%, and 75% quantiles of the ratio between the test risk in the reduced and full bagging settings with uniform and optimized weights $\rho^*_{\text{TND}}$. Results on MUSHROOM and SHUTTLE are left out, as the test risk is 0 in some cases.

Figure I.14: Median, 25%, and 75% quantiles of the ratio between the TND bound in the reduced and full bagging settings with uniform and optimized weights $\rho^*_{\text{TND}}$.

Figure I.15: Test risk, FO, TND and DIS bounds as a function of the fraction $r$ of labeled points. Means and standard deviations over 20 runs are reported.

Table I.2: Values of the computed bounds when using bagging. Mean and standard deviation are reported for 50 runs on each data set. The tightest bound overall is marked in **bold**, with the second tightest marked with underline. Note that the bounds C1 and C2 are only defined for binary data sets.

| Dataset | $\hat{L}(MV, S_{test})$ | FO | C1 | C2 | CTD | TND |
|---|---|---|---|---|---|---|
| ADULT | 0.16941 (0.00303) | **0.45164** (0.00213) | 0.57227 (0.00256) | 0.57318 (0.00312) | 0.88830 (0.00626) | 0.57882 (0.00364) |
| COD-RNA | 0.04018 (0.00150) | 0.18717 (0.00194) | 0.19835 (0.00171) | 0.18054 (0.00181) | 0.20835 (0.00240) | **0.17151** (0.00176) |
| CONNECT-4 | 0.17120 (0.00204) | **0.62706** (0.00148) | - | - | $> 1$ | 0.69258 (0.00230) |
| FASHION-MNIST | 0.11752 (0.00228) | **0.49426** (0.00138) | - | - | 0.81911 (0.00390) | 0.54943 (0.00229) |
| LETTER | 0.03602 (0.00315) | **0.41503** (0.00235) | - | - | 0.83652 (0.00872) | 0.46613 (0.00374) |
| MNIST | 0.03144 (0.00134) | 0.39628 (0.00117) | - | - | 0.51669 (0.00267) | **0.35937** (0.00154) |
| MUSHROOM | 0.00000 (0.00000) | **0.00801** (0.00031) | 0.01638 (0.00061) | 0.04817 (0.00043) | 0.03952 (0.00040) | 0.03539 (0.00034) |
| PENDIGITS | 0.00854 (0.00183) | 0.16515 (0.00181) | - | - | 0.21330 (0.00380) | **0.15211** (0.00238) |
| PHISHING | 0.02916 (0.00356) | **0.13554** (0.00255) | 0.19865 (0.00351) | 0.19105 (0.00415) | 0.22756 (0.00597) | 0.17367 (0.00409) |
| PROTEIN | 0.32959 (0.00500) | $> 1$ | - | - | $> 1$ | $> 1$ |
| SVMGUIDE1 | 0.03129 (0.00628) | **0.15559** (0.00521) | 0.26894 (0.00782) | 0.30776 (0.00823) | 0.42148 (0.01735) | 0.28161 (0.00865) |
| SATIMAGE | 0.08386 (0.00716) | **0.40328** (0.00409) | - | - | $> 1$ | 0.50910 (0.00605) |
| SENSORLESS | 0.00131 (0.00034) | 0.05380 (0.00077) | - | - | 0.03077 (0.00055) | **0.02797** (0.00049) |
| SHUTTLE | 0.00015 (0.00011) | **0.00379** (0.00011) | - | - | 0.00886 (0.00025) | 0.00827 (0.00024) |
| SPLICE | 0.02957 (0.00798) | 0.45351 (0.00822) | 0.55494 (0.00837) | 0.48391 (0.00930) | $> 1$ | **0.43683** (0.00903) |
| USPS | 0.03820 (0.00395) | **0.37237** (0.00289) | - | - | 0.87450 (0.01387) | 0.42214 (0.00463) |
| W1A | 0.01108 (0.00081) | **0.04704** (0.00065) | 0.07233 (0.00096) | 0.07254 (0.00114) | 0.07234 (0.00122) | 0.06649 (0.00111) |

Table I.3: Statistics for each data set when using bagging. We use the following short-hand: $\mathbb{E}_\rho[\hat{L}] = \mathbb{E}_\rho[\hat{L}(h, S_h)]$, $\mathbb{E}_{\rho^2}[\hat{\mathbb{D}}] = \mathbb{E}_{\rho^2}[\hat{\mathbb{D}}(h, h', S_h \cap S_{h'})]$, $\mathbb{E}_{\rho^2}[\hat{L}] = \mathbb{E}_{\rho^2}[\hat{L}(h, h', S_h \cap S_{h'})]$

| Dataset | $\hat{L}(\text{MV}, S_{\text{test}})$ | $\mathbb{E}_\rho[\hat{L}]$ | $\min |S_h|$ | $\mathbb{E}_{\rho^2}[\hat{\mathbb{D}}]$ | $\mathbb{E}_{\rho^2}[\hat{L}]$ | $\min |S_h \cap S_{h'}|$ |
|---|---|---|---|---|---|---|
| ADULT | 0.16941 | 0.20851 | 9459.96 | 0.17422 | 0.12138 | 3359.92 |
| COD-RNA | 0.04018 | 0.08456 | 17348.12 | 0.10315 | 0.03298 | 6228.60 |
| CONNECT-4 | 0.17120 | 0.29985 | 19697.48 | 0.34343 | 0.15527 | 7077.06 |
| FASHION-MNIST | 0.11752 | 0.23465 | 20410.22 | 0.28396 | 0.12142 | 7335.90 |
| LETTER | 0.03602 | 0.18644 | 5785.64 | 0.26280 | 0.08995 | 2032.40 |
| MNIST | 0.03144 | 0.18663 | 20416.56 | 0.27435 | 0.07668 | 7337.68 |
| MUSHROOM | 0.00000 | 0.00019 | 2327.40 | 0.00036 | 0.00001 | 797.00 |
| PENDIGITS | 0.00854 | 0.06403 | 3158.70 | 0.10004 | 0.01828 | 1095.48 |
| PHISHING | 0.02916 | 0.05097 | 3178.92 | 0.05747 | 0.02224 | 1100.30 |
| PROTEIN | 0.32959 | 0.53934 | 7065.98 | 0.59019 | 0.31349 | 2497.42 |
| SVMGUIDE1 | 0.03129 | 0.04606 | 869.08 | 0.04601 | 0.02297 | 284.54 |
| SATIMAGE | 0.08386 | 0.16636 | 1837.52 | 0.19830 | 0.08061 | 623.54 |
| SENSORLESS | 0.00131 | 0.02193 | 17049.78 | 0.03845 | 0.00318 | 6112.32 |
| SHUTTLE | 0.00015 | 0.00069 | 16899.58 | 0.00097 | 0.00023 | 6055.44 |
| SPLICE | 0.02957 | 0.17561 | 895.60 | 0.25165 | 0.04976 | 293.14 |
| USPS | 0.03820 | 0.15736 | 2668.56 | 0.22115 | 0.06944 | 916.28 |
| W1A | 0.01108 | 0.01852 | 14483.72 | 0.01695 | 0.01005 | 5176.26 |

Table I.4: Values of the computed bounds when using reduced bagging. Mean and standard deviation are reported for 50 runs on each data set. The tightest bound overall is marked in **bold**, with the second tightest marked with underline. Note that the bounds C1 and C2 are only defined for binary data sets.

| Dataset | $\hat{L}(MV, S_{test})$ | FO | C1 | C2 | CTD | TND |
|---|---|---|---|---|---|---|
| ADULT | 0.16370 (0.00322) | **0.44940** (0.00193) | 0.53068 (0.00226) | 0.51552 (0.00241) | 0.71364 (0.00431) | 0.51148 (0.00258) |
| COD-RNA | 0.04346 (0.00158) | 0.20793 (0.00194) | 0.19300 (0.00132) | 0.17299 (0.00126) | 0.19433 (0.00161) | **0.16725** (0.00124) |
| CONNECT-4 | 0.17716 (0.00238) | **0.64645** (0.00164) | - | - | >1 | 0.68942 (0.00215) |
| FASHION-MNIST | 0.12258 (0.00244) | **0.50935** (0.00138) | - | - | 0.73495 (0.00307) | 0.54870 (0.00204) |
| LETTER | 0.04326 (0.00374) | **0.47741** (0.00234) | - | - | 0.83563 (0.00740) | 0.51579 (0.00402) |
| MNIST | 0.03553 (0.00153) | 0.43055 (0.00140) | - | - | 0.50697 (0.00288) | **0.38250** (0.00180) |
| MUSHROOM | 0.00000 (0.00000) | **0.00724** (0.00039) | 0.01468 (0.00075) | 0.01959 (0.00036) | 0.01554 (0.00035) | 0.01408 (0.00033) |
| PENDIGITS | 0.01052 (0.00164) | 0.18755 (0.00177) | - | - | 0.18659 (0.00280) | **0.14001** (0.00187) |
| PHISHING | 0.03009 (0.00364) | 0.15753 (0.00222) | 0.19101 (0.00261) | 0.15970 (0.00262) | 0.18902 (0.00357) | **0.14832** (0.00253) |
| PROTEIN | 0.33413 (0.00552) | >1 | - | - | >1 | >1 |
| SVMGUIDE1 | 0.03006 (0.00541) | **0.14470** (0.00359) | 0.22802 (0.00481) | 0.20687 (0.00532) | 0.25874 (0.00852) | 0.18814 (0.00529) |
| SATIMAGE | 0.09063 (0.00713) | **0.40936** (0.00352) | - | - | 0.87653 (0.01369) | 0.45752 (0.00490) |
| SENSORLESS | 0.00187 (0.00038) | 0.06891 (0.00111) | - | - | 0.03011 (0.00048) | **0.02847** (0.00045) |
| SHUTTLE | 0.00024 (0.00011) | **0.00405** (0.00010) | - | - | 0.00551 (0.00017) | 0.00525 (0.00017) |
| SPLICE | 0.03398 (0.00759) | 0.47039 (0.00885) | 0.49763 (0.00868) | 0.38612 (0.00884) | >1 | **0.35799** (0.00857) |
| USPS | 0.04385 (0.00457) | **0.39916** (0.00328) | - | - | 0.72489 (0.01091) | 0.40572 (0.00434) |
| W1A | 0.01121 (0.00068) | **0.04782** (0.00064) | 0.06528 (0.00075) | 0.05949 (0.00079) | 0.05955 (0.00083) | 0.05585 (0.00077) |

Table I.5: Statistics for each data set when using reduced bagging. We use the following short-hand: $\mathbb{E}_\rho[\hat{L}] = \mathbb{E}_\rho[\hat{L}(h, S_h)]$, $\mathbb{E}_{\rho^2}[\hat{\mathbb{D}}] = \mathbb{E}_{\rho^2}[\hat{\mathbb{D}}(h, h', S_h \cap S_{h'})]$, $\mathbb{E}_{\rho^2}[\hat{L}] = \mathbb{E}_{\rho^2}[\hat{L}(h, h', S_h \cap S_{h'})]$

| Dataset | $\hat{L}(\mathrm{MV}, \mathrm{S_{test}})$ | $\mathbb{E}_\rho[\hat{L}]$ | $\min|S_h|$ | $\mathbb{E}_{\rho^2}[\hat{\mathbb{D}}]$ | $\mathbb{E}_{\rho^2}[\hat{L}]$ | $\min|S_h \cap S_{h'}|$ |
|---|---|---|---|---|---|---|
| ADULT | 0.16370 | 0.21105 | 15702.72 | 0.19390 | 0.11409 | 9401.66 |
| COD-RNA | 0.04346 | 0.09649 | 28763.86 | 0.12166 | 0.03566 | 17277.52 |
| CONNECT-4 | 0.17716 | 0.31235 | 32643.02 | 0.35965 | 0.16125 | 19614.80 |
| FASHION-MNIST | 0.12258 | 0.24472 | 33827.88 | 0.29624 | 0.12725 | 20337.04 |
| LETTER | 0.04326 | 0.22119 | 9630.06 | 0.30819 | 0.11160 | 5745.70 |
| MNIST | 0.03553 | 0.20590 | 33828.08 | 0.30029 | 0.08716 | 20331.38 |
| MUSHROOM | 0.00000 | 0.00057 | 3894.98 | 0.00104 | 0.00005 | 2299.78 |
| PENDIGITS | 0.01052 | 0.07817 | 5276.38 | 0.12164 | 0.02289 | 3127.52 |
| PHISHING | 0.03009 | 0.06443 | 5308.52 | 0.07962 | 0.02464 | 3146.54 |
| PROTEIN | 0.33413 | 0.54520 | 11746.54 | 0.59434 | 0.31870 | 7019.70 |
| SVMGUIDE1 | 0.03006 | 0.04796 | 1470.06 | 0.05098 | 0.02247 | 853.08 |
| SATIMAGE | 0.09063 | 0.17665 | 3080.48 | 0.21070 | 0.08664 | 1813.18 |
| SENSORLESS | 0.00187 | 0.03000 | 28265.20 | 0.05216 | 0.00462 | 16971.34 |
| SHUTTLE | 0.00024 | 0.00101 | 28014.66 | 0.00144 | 0.00034 | 16823.72 |
| SPLICE | 0.03398 | 0.19427 | 1511.18 | 0.27746 | 0.05557 | 876.88 |
| USPS | 0.04385 | 0.17617 | 4458.86 | 0.24642 | 0.07926 | 2636.72 |
| W1A | 0.01121 | 0.01991 | 24022.24 | 0.01957 | 0.01013 | 14412.94 |

| Dataset | $\hat{L}(\mathrm{MV}_u, \mathrm{S_{test}})$ | $\hat{L}(\mathrm{MV}_{\rho_{\mathrm{FO}}^*}, \mathrm{S_{test}})$ | $\hat{L}(\mathrm{MV}_{\rho_{\mathrm{TND}}^*}, \mathrm{S_{test}})$ |
|---|---|---|---|
| ADULT | **0.16941** (0.00303) | 0.19136 (0.01335) | 0.17004 (0.00313) |
| COD-RNA | 0.04018 (0.00150) | 0.07193 (0.00530) | **0.03963** (0.00138) |
| CONNECT-4 | **0.17120** (0.00204) | 0.28148 (0.01407) | 0.17123 (0.00202) |
| FASHION-MNIST | **0.11752** (0.00228) | 0.20678 (0.03283) | 0.11895 (0.00222) |
| LETTER | **0.03602** (0.00315) | 0.14998 (0.03493) | 0.03784 (0.00336) |
| MNIST | **0.03144** (0.00134) | 0.16014 (0.03238) | 0.03223 (0.00137) |
| MUSHROOM | **0.00000** (0.00000) | **0.00000** (0.00000) | **0.00000** (0.00000) |
| PENDIGITS | **0.00854** (0.00183) | 0.04752 (0.01515) | 0.00856 (0.00168) |
| PHISHING | **0.02916** (0.00356) | 0.03865 (0.00649) | 0.02935 (0.00355) |
| PROTEIN | **0.32959** (0.00500) | 0.49377 (0.03958) | 0.33402 (0.00578) |
| SVMGUIDE1 | 0.03129 (0.00628) | 0.03786 (0.00764) | **0.03120** (0.00637) |
| SATIMAGE | **0.08386** (0.00716) | 0.13876 (0.02631) | 0.08437 (0.00711) |
| SENSORLESS | 0.00131 (0.00034) | 0.01304 (0.00298) | **0.00118** (0.00029) |
| SHUTTLE | 0.00015 (0.00011) | 0.00022 (0.00015) | **0.00013** (0.00011) |
| SPLICE | **0.02957** (0.00798) | 0.11257 (0.02121) | 0.03005 (0.00769) |
| USPS | **0.03820** (0.00395) | 0.12554 (0.03381) | 0.03954 (0.00417) |
| W1A | 0.01108 (0.00081) | 0.01586 (0.00279) | **0.01106** (0.00081) |

Table I.6: Test risks computed when using different bounds for optimizing $\rho$. Best risk achieved overall is marked in **bold**, while best risk achieved by optimization is marked with underline.

| Dataset | $\hat{L}(\mathrm{MV}_u, \mathrm{S_{test}})$ | $\hat{L}(\mathrm{MV}_{\rho_{\mathrm{FO}}^*}, \mathrm{S_{test}})$ | $\hat{L}(\mathrm{MV}_{\rho_{\mathrm{TND}}^*}, \mathrm{S_{test}})$ |
|---|---|---|---|
| ADULT | **0.16370** (0.00322) | 0.19592 (0.01385) | 0.16427 (0.00337) |
| COD-RNA | 0.04346 (0.00158) | 0.07990 (0.00725) | **0.04282** (0.00167) |
| CONNECT-4 | 0.17716 (0.00238) | 0.29161 (0.01928) | **0.17698** (0.00211) |
| FASHION-MNIST | **0.12258** (0.00244) | 0.23242 (0.01962) | 0.12367 (0.00262) |
| LETTER | **0.04326** (0.00374) | 0.19865 (0.02292) | 0.04613 (0.00341) |
| MNIST | **0.03553** (0.00153) | 0.18514 (0.02914) | 0.03662 (0.00164) |
| MUSHROOM | **0.00000** (0.00000) | **0.00000** (0.00000) | **0.00000** (0.00000) |
| PENDIGITS | **0.01052** (0.00164) | 0.06012 (0.01572) | 0.01070 (0.00174) |
| PHISHING | 0.03009 (0.00364) | 0.05129 (0.00849) | **0.02958** (0.00370) |
| PROTEIN | **0.33413** (0.00552) | 0.51822 (0.02526) | 0.33895 (0.00504) |
| SVMGUIDE1 | **0.03006** (0.00541) | 0.03845 (0.00701) | 0.03126 (0.00532) |
| SATIMAGE | **0.09063** (0.00713) | 0.15094 (0.02514) | 0.09114 (0.00690) |
| SENSORLESS | 0.00187 (0.00038) | 0.01819 (0.00269) | **0.00171** (0.00040) |
| SHUTTLE | 0.00024 (0.00011) | 0.00035 (0.00020) | **0.00016** (0.00012) |
| SPLICE | **0.03398** (0.00759) | 0.12252 (0.02238) | 0.03657 (0.00857) |
| USPS | **0.04385** (0.00457) | 0.14450 (0.02630) | 0.04534 (0.00418) |
| W1A | 0.01121 (0.00068) | 0.01572 (0.00234) | **0.01117** (0.00078) |

Table I.7: Test risks computed when using different bounds for optimizing $\rho$ for random forest trained using reduced bagging. Best risk achieved overall is marked in **bold**, while best risk achieved by optimization is marked with underline.