[Reviews · NeurIPS 2020]

Review 1

Summary and Contributions: This paper presents a PAC-Bayes analysis of the expected risk of weighted majority vote in classification problems. The proposed bound is "second order" in the sense that it is based on the second moment of the losses. Interestingly, the analysis uses a "tandem loss" (new?) which is a loss function over pairs of predictors, the promise is that this tandem loss seems to account for correlations of predictions, and leads to improved bounds. In a specialized version for binary classification, this paper shows how to take advantage of additional unlabeled data for tighter risk estimation. The experiments show that a training method based on this bound allows to improve weights over trees in random forests. These contributions are valuable.

Strengths: Well written paper, clear and rigorous. The analysis of the "tandem loss" is very interesting. I have not seen this kind of loss before in the literature before, I am counting it as part of the novelty of this paper. The proposed bounds based on this tandem loss are new and interesting. The informative experiments validate the theory and are presented/discussed with plenty of detail. Reasonably tight generalization guarantee. Interesting to see that minimization of the bound does not deteriorate predictor performance on the test set. Overall a good paper in my opinion.

Weaknesses: While most of the computed bounds are non-vacuous, they clearly look to be not that tight. Some discussion of this would be valuable. Also a discussion of potential ways to obtain tighter bond values, or whether there is a fundamental limitation.

Correctness: Yes. Yes.

Clarity: This is a positive example of a well written paper.

Relation to Prior Work: Yes.

Reproducibility: Yes

Additional Feedback: In Section 3, regarding definition of tandem loss, it would be helpful to first define (with notation) the tandem loss for a pair (h,h') and a given labelled example (X,Y), and then define L(h,h') as expectation of that tandem loss function over a random (X,Y). In section 3.1, when defining the expected disagreement of a pair (h,h'), the expectation needs only the distribution of X (not of Y) and this will be better aligned with the thought that extra unlabelled data is available (for which integrating over the distribution of Y makes no sense). In Section 3.3, for convenience, it would be helpful to reproduce here the previous C-bound(s). Then one will be able to contrast with Theorem 6 more easily. Line 200: "The set S' may overlap with the inputs X of the labeled set S, however, S' may include..." Line 239: "used also by McAllester [2003]" Figure 1: if possible (i think it is) show the value of the test risk (black lines) on the vertical scale. Similar for corresponding figures in the appendix. Line 471: should it be " \hat{\ell}_i \hat{\ell}_i^T " ? *** AFTER AUTHOR RESPONSE *** Thank you for taking the time to respond to my comments.


Review 2

Summary and Contributions: The classical PAC-Bayesian generalization bounds over the majority vote classification suffers from the fact that a bound on the randomized classifier loss (sometimes called the Gibbs risk) required to be multiplied by a factor 2 in order to be turned into the majority vote risk. The authors proposed a new strategy to bound the majority vote risk based on a simple but clever inequality that uses the randomized predictor second order information. It turns out to be a simpler strategy than previous similar attempt proposed by Lacasse et al. (2007) & Germain et al. (2015) - the so-called C-bound -, and empirical experiments shows that it leads to tighter majority vote bounds.

Strengths: The paper is well written. It builds on previous works in a sound manner. Thus, the results (both theoretical and empirical) are not ground breaking, but they are well motivated and discussed. Overall, this study provides ideas and insights that enrich the PAC-Bayes framework and can be used in by others in future works.

Weaknesses: The authors properly mention that their work build on (among others) Lacasse et al. (2007) & Germain et al. (2015) work on PAC-Bayes and majority votes. However, it should be mentioned that: - The proposed "tandem loss" is a multiclass generalization of the "joint error" of Lacasse et al. & Germain et al. - Lemmas 2 and 4 are actually stated in Lacasse et al (2006) as Equations 7 and 8. - In the experiment section, the empirical bound values C1 and C2 are quickly presented as "the two forms of C-bounds of Germain et al." For completeness, these forms should be explicitly stated. - Note that the "second C-bound form" allowed Germain et al. to bound both the expected first order loss and the disagreement simultaneously. The same strategy can be adapted to the proposed Theorem 10.

Correctness: Both theoretical results and experiments are sound. The completeness of the supplementary material is worth mentioning, as it provides clear proofs, complementary discussion, extra experiments, and source code to reproduce the experiments,

Clarity: The paper is very well written and pleasant to read.

Relation to Prior Work: The paper properly discusses prior works (Germain et al. 2015, Thiemann et al. 2017, Lorenzen et al. 2019), except for the few points mentioned as "weaknesses" above.

Reproducibility: Yes

Additional Feedback: *** AFTER AUTHOR RESPONSE *** The authors provide an honest rebuttal. I agree with points (*) and (**). My opinion remains unchanged: This is a great paper!


Review 3

Summary and Contributions: This paper derives a new risk upper bound for the weighted majority vote classifier in the multi-class classification setting. This bound is based on the tandem loss notion that is the covariance between the errors of two hypotheses. The authors show that under some conditions, the bound can be tighter than the classical first-order bound (FO), which was validated by empirical results. To practically compute the bound, they derive two corresponding PAC-Bayes bounds, the classical one and one for optimization. In addition, they show how unlabeled data can be incorporated in the bound estimation for the binary case. Finally, they show that their bound (TND) is a better optimization criterion than FO to find an optimal posterior over the hypothesis space.

Strengths: 1) The authors provide a thorough analysis of their bound and clearly indicate the pros and cons of using it. 2) Although the proposed bound is simpler than the C-bound, it is more advantageous from the point of view of estimation. In addition, the C-bound is not differentiable in the general case. 3) There is a particular interest of the NeurIPS community in the learning theory. This work matches this interest, especially, given the fact that there are no many works considering the multi-class classification framework.

Weaknesses: - In the experiments, the authors have compared with the C-bound only in the binary case, while it exists the multi-class version [1]. It is different from the C-tandem bound, which is a relaxed version of it. - The experiment with the posterior optimization does not seem promising. In this task, we would be more interested in the improvement over the test error rather than the bound value. As one can see, the posterior found by minimization of TND does not yield a better solution in terms of the test error, even the bound becomes tighter. In addition, in this classical supervised setting the cross-validation will do the job better, so there is no necessity of using bounds for the optimal posterior search task.

Correctness: 1) The empirical methodology seems correct. 2) It seems there is an error in the proof of Lemma C.14. Line 436, transition from 2nd to 3rd row: the authors cut by E[X]^2 on the left side, but cut by E[X^2] on the right side, where the sign of the other multiplier is not known. Hence, it has to be re-proven. Since this lemma is not one of the main results, I do not change my grade. In the other proofs I have not found any error.

Clarity: The paper is well written and contains all details for the reproducibility of the results. However, for readers not acquainted with the domain it would be hard to follow. Probably, some more interpretation could be added in Section 2.

Relation to Prior Work: The prior work is thoroughly cited and the difference from previous contributions is clearly discussed. One remark: since the paper is devoted to the multi-class case, the authors have to also cite other results for the majority vote in this framework [2,3] and discuss the difference.

Reproducibility: Yes

Additional Feedback: Below one can find my questions and suggestions for authors. I rated this article as 7 only on the condition that my questions / suggestions will be treated by the authors. 1) Please, could you improve the prior work discussion and compute the multi-class C-bound [1] for the multi-class experiments? 2) Please, re-prove Lemma C.14. 3) I would suggest to move the posterior optimization experiment to the supplementary and insert some other results instead (maybe Fig H.15?) 4) Line 281: How did you do this for the C-tandem bound, if it has L(h,h') in the denominator and -L(h) in the numerator? References [1] Laviolette, François, et al. "Risk upper bounds for general ensemble methods with an application to multiclass classification." Neurocomputing 219 (2017): 15-25. [2] Morvant, Emilie, Sokol Koço, and Liva Ralaivola. "PAC-Bayesian generalization bound on confusion matrix for multi-class classification." Proceedings of the 29th International Coference on International Conference on Machine Learning. 2012. [3] Feofanov, Vasilii, Emilie Devijver, and Massih-Reza Amini. "Transductive Bounds for the Multi-Class Majority Vote Classifier." Proceedings of the AAAI Conference on Artificial Intelligence. Vol. 33. 2019. ============================================================== UPDATE: I thank the authors for providing the reply. I agree with their points. ==============================================================


Review 4

Summary and Contributions: The paper presents two PAC-Bayes bounds aimed at the study of ensemble classifiers, since they take into account the existence of correlations between the prediction errors of the different members of the ensemble. For both bounds the paper provides a general oracle version and an empirical, PAC-Bayes bound that can be used in real data problems. The tightness of the bounds is extensively evaluated in a complete experimental work.

Strengths: The theoretical analysis is very good and easy to follow (proofs are well detailed in the appendix). The empirical evaluation is very complete. The discussion perfectly details in which cases the proposed second order (use correlations among members of the ensemble) bounds are a better option than the usual first order ones. I think the paper is very relevant since ensemble methods (like random forest) are very used among practitioners and sometimes it is difficult to find attempts to analyze theoretically their good performance.

Weaknesses: The appendix includes a lot of information that is treated with small detail in the main body, specially in the experimental section. Besides, and thinking in the broader audience of the paper, the results achieved by the PAC-Bayes bounds, although very tight, still have room for improvement compared to the test risk.

Correctness: I think the theoretical analysis is correct, I checked more or less most of the proofs and I think they are correct. The empirical methodology is explained in large detail (in the appendix) and I think it is correct. Besides, I also think it is easy to reproduce. Therefore in my view the claims are well supported.

Clarity: The paper is clearly written and easy to read. Minor details: - line 95: I think there should be another indicator function inside the expectation wrt \rho. Also in the same place in lines 97 and 98, in order to transform the boolean nature of h(X) not_equal Y in a number - eq under line 109. I found the notation for the square of the expectation a bit confusing, checking the appendix one easily notices that you mean (E(x))^2 and not E(x^2), but in the main body it is not so clear (in my view)

Relation to Prior Work: I think the paper describes with good detail the related work, and makes good and also detailed comparisons to point out what are the contributions of this paper and which are the novel aspects of these contributions.

Reproducibility: Yes

Additional Feedback: -------------------------------------------------------------------- Update after rebuttal: I thank the authors for their response --------------------------------------------------------------------

[Author Response · NeurIPS 2020]

We thank all the reviewers for their insightful comments, suggestions, and references.

**Reviewer 1:**

*Novelty of tandem loss:* it is not new, but we were not aware of the prior work, we thank Reviewer 2 for bringing it up.

*While most of the computed bounds are non-vacuous, they look to be not that tight. Some discussion of this would be*
*valuable. Also a discussion of potential ways to obtain tighter bond values, or whether there is a fundamental limitation.*

We provide some discussion in Sections 3.2 and 4.4. The major challenge is the estimation of the tandem loss, which is
based on overlaps of OOB samples, which are small [see Section 4.4]. It could be that a better estimation technique
could be designed in the future. Another limitation is the oracle bound. For example, in the independent case with the
growth of the number of classifiers it converges to $4L(h)^2$, whereas $L(\mathrm{MV}_\rho)$ converges to zero [see Section 3.2].

**(\*)** *Reproduction of C-bounds:* Reproduction of C-bounds requires definition of a margin. We believe that adding it to
the body might divert the attention from the main thread of the paper, but we will be very glad to add a section in the
appendix, where we introduce the margin, discuss the relation with the tandem loss, and provide the C-bounds.

*Line 471:* Yes, you are right, thank you!

**Reviewer 2:**

Thank you for providing references to "joint error" and Equations 7 and 8 in Lacasse et al., we will add them.

*Reproduction of C-bounds:* see our reply **(\*)** to Reviewer 1.

**(\*\*)** *Adaptation of the strategy from the "second C-bound form":* Note that in our application the first order loss and the
disagreements are estimated on different subsamples. The first order loss is estimated on OOB samples, whereas the
disagreements are estimated on overlaps of OOB samples and unlabeled data when available. It is not immediately
clear whether joint estimation would give an advantage, we will look at it in future work.

**Reviewer 3:**

*Posterior optimization*

We agree that posterior optimization did not improve the test error. However, we note that in prior work posterior
optimization was either impossible (in C-bounds, except for highly limiting cases of aligned posteriors in binary
classification) or led to considerable deterioration of the test error (as we demonstrate for the first order bound).
Therefore, we see absence of deterioration of the test error as a step forward relative to prior work.

*1) compute the multi-class C-bound*

The multi-class C-bound based on the $w$-margin with $w = 1/2$ (Corollary 1 of Laviolette et al.) is equivalent to our
oracle C-tandem bound in Theorem 6. The values of empirical C-tandem bound are reported in the paper. We note that
there may be multiple ways of going from the oracle to an empirical bound, but not all of them are directly applicable in
the OOB setting, see our reply **(\*\*)** to Reviewer 2. We also note that the general multi-class C-bound (Theorem 2 of
Laviolette et al.) cannot be evaluated of the OOB setting, because the max operator in their definition of the margin in
equation (3) cannot be exchanged with expectation and the estimation cannot be done using pairs of hypotheses.

*2) Please, re-prove Lemma C.14.*

Oh, sorry, we missed that the square was outside the expectation on the left and inside on the right when we canceled
the terms. Thanks for catching it! The fix is easy. Instead of canceling, take $-\mathbb{E}\left[X\right]^2 \varepsilon^2$ to the right hand side. Then
$\mathbb{E}\left[X\right]^2 \varepsilon^2 - 2\varepsilon \mathbb{E}\left[X\right]\mathbb{E}\left[X^2\right] + \mathbb{E}\left[X^2\right]^2 = (\mathbb{E}\left[X\right]\varepsilon - \mathbb{E}\left[X^2\right])^2 \geq 0.$

*3) move the posterior optimization experiment to the supplementary and insert some other results instead (Fig H.15?)*

We will have an extra page if accepted, so we can have both in the body.

*4) Line 281*

The kl-inequality has an upper and a lower inverse, which give an upper and a lower bound, respectively. We have used
the lower bound.

Thanks a lot for the references to additional work on multiclass classification!

**Reviewer 4:**

*lines 95, 97, and 98:* You are right, thanks!

*line 109:* We will add extra brackets, thanks!

[Meta-Review · NeurIPS 2020]

The authors proposes a new PAC-Bayes bound on majority vote classifiers (such as, but not restricted to, classifiers that comes from ensemble methods). Contrarily to most of known PAC-Bayes bounds, their proposed bound do not only take into account the so called Gibbs risk (which can be view as the average quality of the voters one which the majority vote is build), but also on some second order information (that quantifies how much the classifiers of the majority votes are decorrelated on their errors. A similar attempt has been proposed by Lacasse et al. (2007) & Germain et al. (2015). Their approach is nevertheless more general, simpler and seems to leads to tighter majority vote bounds. Most of the learning algorithms can be analysed through the PAC-Bayesian theory. However, classical PAC-Bayesian generalization bounds for classification suffers from the fact that the bound rely on the first moment only. The approach of Lacasse et al. (2007) & Germain et al. (2015) being until now the exception. However, on most cases the proposed bounds happened to be too looses. The proposed bound in this paper seems to overcomes this weakness and gives also a more precise understanding of learning algorithms that outcome majority-vote-ish classifier. On my opinion, this results is of high interest for the Machine Learning Community.